

# INGe: Intensity-ground motion dataset for Italy

Ilaria Oliveti[1], Licia Faenza[2], and Alberto Michelini[1]

[1]Istituto Nazionale di Geofisica e Vulcanologia, sezione ONT, Rome, Italy.
[2]Istituto Nazionale di Geofisica e Vulcanologia, sezione ONT, Bologna, Italy.

**Correspondence:** Ilaria Oliveti (ilaria.oliveti@ingv.it)

**Abstract.** In this paper we present an updated and homogeneous earthquake data set for Italy compiled by joining the Italian Macroseismic Database DBMI15 and the Engineering Strong-Motion (ESM) accelerometric data bank. The database has been compiled through a extensive procedure of selection and revision based on two main steps: 1) the removal of several earthquakes in DBMI15 because the data source has been considered to be largely unreliable and 2) the extraction of all the localities reporting intensity data which are located within 3 km from the accelerograph stations that recorded the data. The final data set includes 323 recordings from 65 earthquakes and 227 stations in the time span 1972-2016. The events are characterized by magnitudes in the range 4.0-6.9 and depths in the range 0.3-45.0 km. Here, we illustrate the data collection and the properties of the database in terms of recording, event and station distributions as well as Mercalli-Cancani-Sieberg (MCS) macroseismic intensity points. Furthermore, we discuss the most relevant features of engineering interest showing several statistics with reference to the most significant metadata (such as moment magnitude, several distance metrics, style of faulting etc). The data set can be downloaded from data repository Zenodo at https://doi.org/10.13127/inge.1 (Oliveti et al., 2020).

## 1   Introduction

Availability of public, high-quality and verified data sets is important in order to develop new techniques and for bench-marking and comparing the performance of existing ones. These data sets provide (i.) a common ground upon which it is possible to make fair comparisons between different techniques and (ii.) their availability can save much time to the developers of new methodologies.

In recent years in seismology, much attention has been put to the development of techniques that adopt macroseismic data from databases (e.g., DBMI15: https://emidius.mi.ingv.it/CPTI15-DBMI15; SISFRANCE: BRGM-EDF-IRSN; https://www.sisfrance.net/) or gathered rapidly after an earthquake [e.g., internet-based questionnaires, such as the U.S. Geological Survey (USGS) "Did You Feel It?" (DYFI) system (Wald et al., 2012) and the Italian "Hai Sentito Il Terremoto?" HSIT database (Tosi et al., 2007); the LastQuake smartphone app developed by the European Mediterranean Seismological Centre (EMSC) for global earthquake eyewitnesses (Bossu et al., 2018)] to quantify the strong ground motion and the associated impact. To this end, there have been developed several ground motion intensity conversion equations (GMICE) that allow to convert from a given intensity scale (e.g., the modified Mercalli intensity scale (MM or MMI), the Mercalli-Cancani-Sieberg scale (MCS) and the European macroseismic scale (EMS)) to ground motion units and vice-versa. It follows that intensity data, despite





their sometime inevitable subjectivity (Musson et al., 2010), can play an important role when compiling maps of seismic hazard that avail of historical earthquakes that have no or very few recorded data, or to increase the density of the observations when producing shakemaps right after an earthquake through the compilation of internet questionnaires. In particular, in the

seismological-engineering community there have been much effort to assemble two types of databases — peak ground motion (PGM) parameters extracted from instrumental recordings and macroseismic data obtained through bibliographic research of historical sources or, for the recent events, from questionnaire based methodologies.

In the first case, comprehensive sets of event, source and station metadata, in addition to the PGM data and other parameters of interest, are usually summarized in a flatfile and used for calibrating Ground Motion Models (GMMs), Probabilistic Seismic

Hazard Assessment (PSHA), calibrating ShakeMap local or regional configurations, and for other engineering applications. Several researchers have presented such databases for different parts of the world (e.g., Chiou et al. (2008) for shallow crustal earthquakes as part of the NGA-West 1 project; Akkar et al. (2010) for Turkey; Arango et al. (2011b) for the Central American subduction zone; Arango et al. (2011a) for the Peru-Chile subduction zone; Pacor et al. (2011) for the ITACA database in Italy). Recently, the ESM flatfile (Lanzano et al., 2018, 2019) updates and greatly enrich with new data existing European data

sets, such as ISESD (Internet-Site for European strong motion Data (Ambraseys et al., 2004)) and RESORCE database (Akkar et al., 2014).

With regard to macroseismic databases, a trans-national European database called AHEAD (Archive of Historical EArthquake Data, (Locati et al., 2014)) has been created supporting the growth of other European intensity databases (Catalonia, Spain, Portugal, Greece and UK) while, at world-wide scale, the Global Historical Earthquake Archive, GHEA (Albini et al.,

2014) has been compiled. This latter archive collects and critically organises the best and most recent information available for earthquakes falling within the time-window 1000-1903 and with magnitudes equal to or higher than 7. In Italy, the latest version of the Italian Macroseismic Database, DBMI15 (Locati et al., 2019), has been released in July 2016, replacing the previous version, called DBMI11 (Locati et al., 2011). DBMI makes available a set of macroseismic intensity data related to Italian earthquakes that covers the time window 1000-2016. Intensity data derive from studies by authors from various Institutions,

both in Italy and bordering countries (France, Austria, Slovenia, and Croatia).

The objective of this work is to develop a benchmark data set upon which existing and novel techniques can be validated and compared. The starting point of our work has been the data set provided by Faenza and Michelini (2010) and Faenza and Michelini (2011). In these works, the authors assembled, respectively, a Peak Ground Acceleration (PGA) and Velocity (PGV)/intensity data set and a Spectral Acceleration (SA)/intensity data set. They adopted the DBMI04 intensity database

(Stucchi et al., 2007), a previous version of the Italian macroseismic database, and the above mentioned ITACA accelerometric data bank (Pacor et al., 2011). Here, we merged the latest versions of these sources of data with the inclusion of PGA, PGV and SA values (at 0.3, 1.0, and 3.0 s), we add more recent events and remove several earthquakes whose data were proved to be unreliable.

The article is organized as follows. In section 2, we describe the data collection and selection procedure adopted for the

compilation of the data set in terms of the primary sources of information. In section 3, we discuss the main database fields including their geographical, temporal, and magnitude distribution. This section presents also some general statistics of the data





set, with reference to the most significant metadata for GMMs calibrations, such as moment magnitude, focal depth, several
distance metrics, style of faulting and parameters for site characterization, as well as the distribution of intensity measures. The
paper ends with a brief discussion of the database and its potential applications, but also looking to how this resource may be
updated and improved in the future.

## 2   Compilation of the data set

The ESM flatfile (Lanzano et al., 2018) is a parametric table which contains strong motion data and associated reliable meta-
data of manually processed waveforms related to the ESM database. This flatfile was built within the Thematic Core Service
for Seismology of the project EPOS (European Plate Observing System, .epos-eu.org). The data set for flatfile compilation
includes 2,179 earthquakes recorded by 2,080 stations from Europe and Middle-East during 1969-2016 and originated in
different tectonic environments, whose magnitude ranges from 4.0 to 8.0. The magnitude is computed from moment tensor
solutions provided by different sources [e.g., the Regional Centroid Moment Tensor, RCMT, and the Time Domain Moment
Tensor, TDMT, catalogues (Pondrelli et al., 2006; Scognamiglio et al., 2009) selected according to an hierarchical schema that
firstly privileges earthquake-specific studies, secondarily the Moment Tensor solution (Ekström et al., 2012) and, finally, the
regional or international bulletins, such as the International Seismological Centre (ISC)]. Strong motion intensity measures
consist of peak and integral parameters and duration of each waveform. The periods at which the spectral amplitudes of the
(5% damping) acceleration and displacement response are computed range 0.01-10 s, whereas the amplitudes of the Fourier
spectrum for the frequency range 0.04–50 Hz. The site classification is based on the average shear wave velocity in the upper-
most 30 meters ($V_{S30}$), according to the Eurocode 8 (Code (2005)) categorization scheme. $V_{S30}$ values are obtained from in
situ geophysical measurements, where available, or derived from geology maps. In addition, an estimation of $V_{S30}$ is provided
using the empirical correlation with the topographic slope by Wald and Allen (2007). Furthermore, the flatfile includes the
epicentral distance for all the records and, when the fault geometry is available, the Joyner-Boore distance. More details on
the structure and organisation of the flatfile are discussed by Lanzano et al. (2019). To compile our database, we have ex-
tracted from the ESM flatfile the event and station information, magnitude estimates, distance measurements, style of faulting,
maximum among the two horizontal components of peak ground acceleration (PGA), peak ground velocity (PGV), and peak
response spectral acceleration amplitudes (at 0.3, 1.0, and 3.0 s). The choice of the periods is based on those used in ShakeMap
(Wald et al., 1999; Worden et al., 2020). The latest version of the Italian Macroseismic Database, DBMI15 (Locati et al., 2019),
includes 122,701 Macroseismic Data Points (MDPs) related to 3,212 earthquakes. DBMI uses a specific and continually up-
dated gazetteer related to the whole Italian territory. In the gazetteer each record is associated to a locality, with place name,
an identifier, and other useful information. As explained in Locati et al. (2019), the term "locality" equally refers to either
region, province, or county capitals, and to variously sized hamlets, towns, or cities. The gazetteer ensures the correspondence
between the place name of a locality and a pair of geographical coordinates matching the average macroseismic intensity value
of a more or less large area with a point. Sometimes the available data are not detailed enough to evaluate the intensity with
a high degree of validity, and such an uncertainty is represented with a range (e.g. 6.0-7.0, 7.0-8.0); in this case, we adopted





the average value of the range. However, in other cases the uncertainty is considered so high that DBMI15 adopts one of the available non-conventional descriptive values (e.g. "HD", "D", or "F"). In particular, "F" correspond to class 4.0, "HF" correspond to class 5.0; "SD" correspond to class 5.5; "D" correspond to class 6.5; "HD" correspond to class 8.5, as described in Locati et al. (2019). MDPs collected and organized in DBMI15 come from studies of different authors and institutions, such as Macroseismic Bulletin, online databases (e.g., CFTI4Med; Guidoboni et al., 2007 and reports specifically prepared

for updating the data set content. To generate our dataset, we have extracted all the MDPs from DBMI15 corresponding to earthquakes listed in the ESM flatfile and that have not been listed in the Macroseismic Bulletin since this latter data source has been proved to be largely unreliable. Then, we have made a cross-matching between ESM and DBMI15 data sets. In order to pair intensity and PGM values, we have chosen a distance criteria, common to most of the studies in this field (e.g., Caprio et al., 2015; Locati et al., 2017) whereas, recently, Gomez-Capera et al. (2020), to warrant similarity in terms of site response,

correlated these observations keeping a threshold distance of about 3 km and also through the check of geological and topographic conditions match. In our work we have selected the localities reporting intensity data which are located within 3 km from the strong motion stations that recorded the data. In some cases we have found that the same recording stations could to be paired to different intensity points and, according to our selection criteria, we have chosen the closest ones. Noteworthy and in addition to the previous data sets compiled by Faenza and Michelini (2010) and Faenza and Michelini (2011), this new

assembled data set includes intensity-PGM pairs at intensity levels larger than 8.0, thanks to the inclusion of recent earthquake data.

## 3   Data and metadata

The information that was included for the characterisation of each data point in the dataset can be summarised as follows:

- **Earthquake source parameters**: primary ESM and INGV event ids, date and time of occurrence (origin time), hypocen-
tral coordinates (geographical coordinates and depth), style of faulting (SoF), magnitude (moment-Mw or local-ML). The moment magnitude is available for 88% of the data. Local magnitude ML is used when Mw is not provided. In the following, we will refer to a generic "Magnitude" or "M" matching either Mw or ML according to the above described procedure;

- **Station information**: network and station code, and location of the receiver; EC8 class, measured and calculated $V_{S30}$
from the ESM flatfile, and extracted $V_{S30}$ from the $V_{S30}$ grid adopted by ShakeMap (Michelini et al., 2020);

- **Distance measurements**: epicentral distance, $R_{EPI}$, azimuth and finite-source distance measure related to fault geometry $R_{JB}$, distance between the selected MCS points and the strong motion stations. The Joyner-Boore distance is available for 60% data. Epicentral distance is used when $R_{JB}$) is not provided. In the following, we will refer to a generic "Distance" matching either $R_{JB}$ or $R_{EPI}$ according to the above described procedure.

- **Peak ground motion values**: the maximum between the two horizontal components of the peak ground motion measures (PGA and PGV) and the 5% damping elastic response spectral ordinates in acceleration (SA) at 0.3, 1.0 and 3.0 s;





– **macroseismic data**: MCS values located within 3 km from the stations (referred to a locality, with place identifier and name and a pair of geographical coordinates) and extracted $V_{S30}$ from the $V_{S30}$ grid adopted by ShakeMap (Michelini et al., 2020).

Moving to the description of the data taken from DBMI15, first of all we note that our data set, which contains 323 macroseismic data points related to 65 earthquakes and 204 localities, shows that the higher MCS intensities correspond to the previously mentioned strong seismic sequences occurred in Italy: the 1976 Friuli, the 1980 Irpinia, the 2009 Abruzzo and the 2016–2017 Central Italy (Fig. 1). The number of available MDPs per earthquake is extremely variable; 40% of earthquakes have only 1 MDP; about 11% of earthquakes have 2 MDPs; about 39% have a set of MDPs between 3 and 10 MDPs; a seventh
has between 10 and 50 MDPs; only one has more than 50 MDPs (Fig. 2). No spatial cluster of earthquakes by number of MDPs is observed as they are equally distributed all along the Italian peninsula, while it is evident that earthquakes with a large number of MDPs occurred in 2016 (Figs. 1 and 2).

Our dataset includes 65 earthquakes (Fig. 3) that occurred between June 1972 and October 2016. The earthquakes in the data set were recorded by 227 stations (Fig. 4) located at distances within 300 km from the earthquakes. The epicentral locations
showed in Fig. 4 shows that the geographical distribution of the events reflects the pattern of Italian seismicity.

The distribution of MDPs in km distance and $V_{S30}$ is illustrated in Fig. 5, where site-to-MDP distance is defined as the closest distance between the MDPs and the strong motion stations that recorded the data. As shown, the site-to-MDP distances range from 0.013 km to 2.938 km. The $V_{S30}$ values are extracted from the $V_{S30}$ grid adopted by ShakeMap (Michelini et al., 2020). This figure evidences that most intensity-PGM pairs do not distance more that 1 km and are rather homogeneously
distributed in terms of $V_{S30}$.

Relative position between earthquakes and recording stations are expresses through the event-to-station azimuth (degrees) and the distance (km). The event distribution in km distance and azimuth is reported in Fig. 6. The data set doesn't contain earthquake signals arriving at receiver from all azimuths; there are gaps in the azimuthal coverage along the axis north-east south-west at larger distances. This is mainly due to the geographical setting of the Italian peninsula. The distribution of the
source to site distance in km is given in Fig. 7. We observe that most of the recordings were acquired within 80 km from the epicenter. In Fig. 7 $R_{JB}$ if available otherwise $R_{EPI}$.

Fig. 8 shows the number of events in the database as a function of time for different magnitude ranges. A larger number of earthquakes is observed in those years when important sequences occurred (i.e., Friuli 1976; Irpinia, 1976; L'Aquila, 2009; Emilia, 2012 and Central Italy, 2016).
Fig. 9 illustrates the histograms of (a) magnitude, (b) focal depth and (c) SoF. The most of the data are available at magnitudes about 4.0 to 5.0, underlining the dominance of moderate events (Fig. 9a). A good part of the data are also in the magnitude range 6.0-7.0, due to the contribution of the previously mentioned events. The following features of the events are also considered: focal depth and focal mechanisms. The distribution of earthquakes focal depths (Fig. **??**b) indicates that seismicity is concentrated in the upper 30 km of the crust, corresponding to about 94% of the total events. Looking at the comparison in
terms of focal mechanisms in Fig. 9c, the Normal Faulting (NF) earthquakes are prevalent (40%). Conversely, 20% and 17% of the total events has Thrust (TF) and strike-slip (SS) style of faulting, respectively.





The magnitude-distance distribution of our data set is given in Fig. 10, grouped by style of faulting. The Joyner-Boore distance ($R_{JB}$) is relevant only for events with M > 5.5 and is available for 194 records. Data are quite well sampled for distance between 10 and 100 km. Looking at the focal mechanisms distribution in Fig. 10, the normal faulting style is predominant for
strong events with magnitude comprises between 6.0 and 6.9.

The histograms of the data points with reference to the EC8 site class (Code (2005)) derived from different $V_{S30}$ sources at all stations are shown in Fig. 11. Direct average shear wave velocity $V_{S30}$ values obtained from geophysical investigations are available for 89 stations, corresponding to 39% of the recording sites (dark gray bars in Fig. 11). In the other cases, the site category was estimated on the basis of the $V_{S30}$ calculated from topographic slope (light gray bars in Fig. 11) according
to Wald and Allen (2007) or on the basis of the $V_{S30}$ extracted from the $V_{S30}$ grid adopted by ShakeMap (Michelini et al., 2020) (smokewhite bars in Fig. 11). Apart from the unreliable site classification determined from the incomplete $V_{S30}$ measured data set, the majority of the recording stations are classified as class EC8-B in the both remaining cases (about 58% and 76% ,respectively). Instead, the second numerous class is EC8-A (31%) for calculated $V_{S30}$ and EC8-C (14%) for extracted $V_{S30}$ data points. No stations are classified as EC8-D and EC8-E (Fig. 11).

Fig. 12 shows the distribution of the strong motion and MCS intensity data versus distance grouped by style of faulting. Overall, the database is quite well distributed although we note that only two data-points are related to stations with distances > 200km and there are few intensity data at closer distances for small intensity values (i.e. in the range $3 \leq$ MCS $\leq 3.5$). This follows from the DBMI15 data being compiled for damaging events (i.e. medium-large magnitude earthquakes producing macroseismic damage; e.g. Allen and Wald, 2009). Also the removal of several earthquakes, whose source has proved to be
largely unreliable, affects the number of intensity data when the distance is very low. However, Fig. 12 also illustrates the relevant number of MDPs with moderate intensities, and in particular between 4 and 5. The increase, in comparison to the previous DBMI releases, is due to the inclusion of many moderate energy earthquakes, in particular after the 19th century (Locati et al., 2019). Looking at the focal mechanisms distribution in Fig. 12, the normal faulting style is predominant for high peak ground motion values.

## 4   Conclusions

A new Italian dataset has been compiled comprising 65 events of magnitude 4.2–6.9 that occurred from the year 1972 through 2016 for a total of 323 pairs of macroseismic and ground motion parameters. The data set can be used as reference to benchmark studies seeking correlations between ground motion parameters and MCS macroseismic intensities.

Much effort has been invested in its compilation to identify the earthquakes to be included with the goal of providing an
updated and more reliable version of the data set compiled initially by Faenza and Michelini (2010) and Faenza and Michelini (2011). The work required (i) the intersection of two different sources, the DBMI15 intensity database (Locati et al., 2019) and the ESM accelerometric data bank (Lanzano et al., 2018), (ii) the removal of several earthquakes because the data source was considered unreliable and (iii) the selection of only the closest localities reporting intensity data which are located within 3 km from the recording stations.





In addition to the PGM and macroseismic intensity data pairs, each datum includes earthquake information (e.g. origin time, depth, magnitude, magnitude type, focal mechanism, etc), and recording station information (e.g. station code and location of the receiver, EC8 site class attribution, $V_{S30}$ values), and distance measurements.

The data collected can be used for development and testing of Ground Motion Intensity Conversion Equations (GMICE) and Intensity Prediction Equations (IPE). These both are important for seismic hazard studies and for the calculation of ShakeMaps.

Overall, the publication of this data set is expected to promote the adoption of best practices and to accelerate research progress.

*Data availability.* The dataset compiled in this study is based on two primary databases: the ESM accelerometric data bank available at https://esm.mi.ingv.it/flatfile-2018/ and the DBMI15 intensity database available at https://emidius.mi.ingv.it/CPTI15-DBMI15. $V_{S30}$ values were derived from the $V_{S30}$ grid adopted by ShakeMap (Michelini et al., 2020). Assembled data set table (csv format) may be found at

https://doi.org/10.13127/inge.1 (Oliveti et al., 2020)

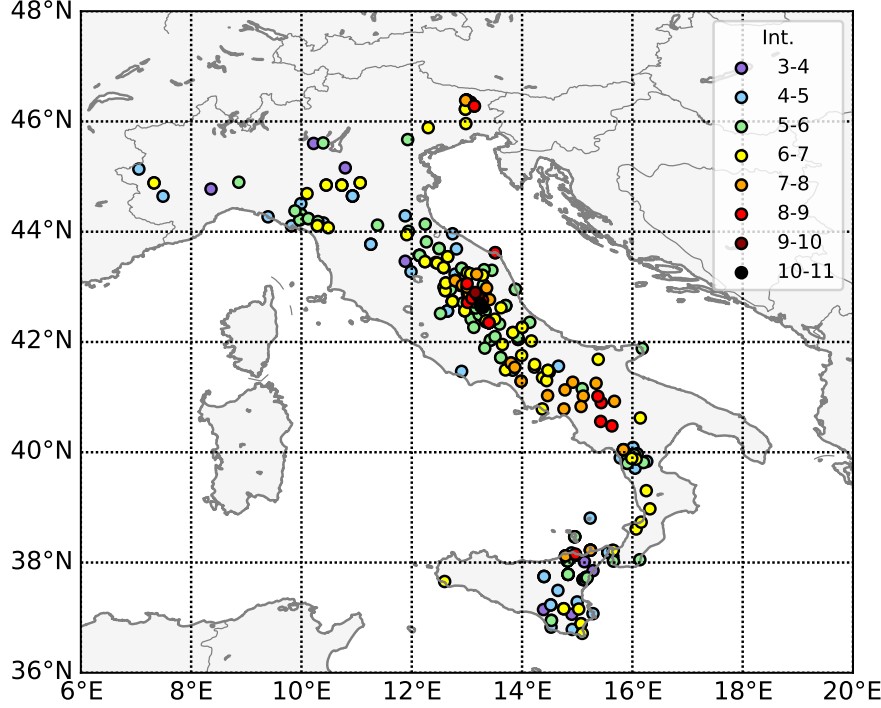

**Figure 1.** The 323 selected observed MCS intensities for the earthquakes included in our data set

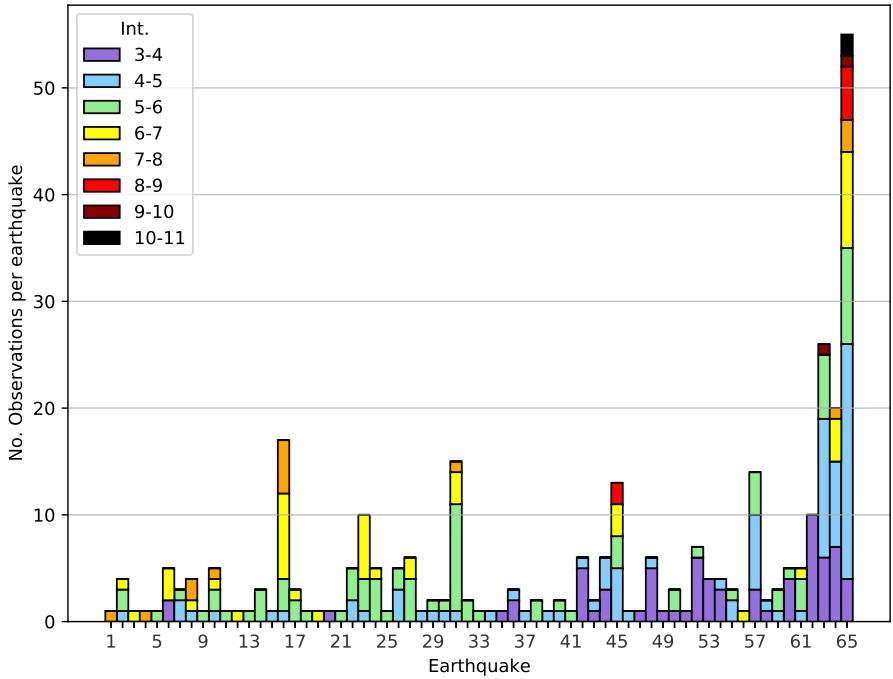

**Figure 2.** Number of MDPs per earthquake for different MCS intensities ranges. The earthquakes are sorted in chronological order (from 1972 to 2016)





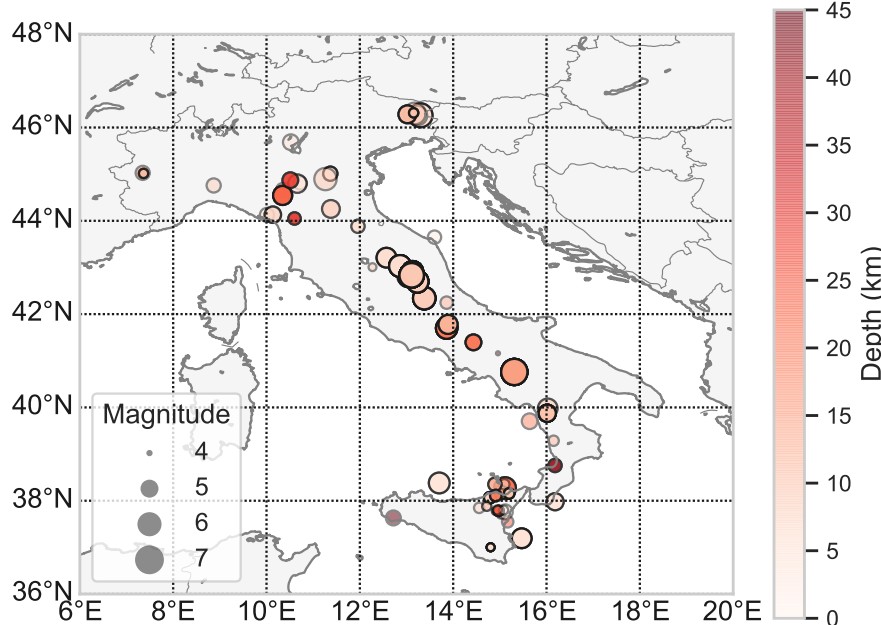

**Figure 3.** Epicentral map of the earthquakes in the dataset. Sphere sizes were plotted relative to their magnitude value

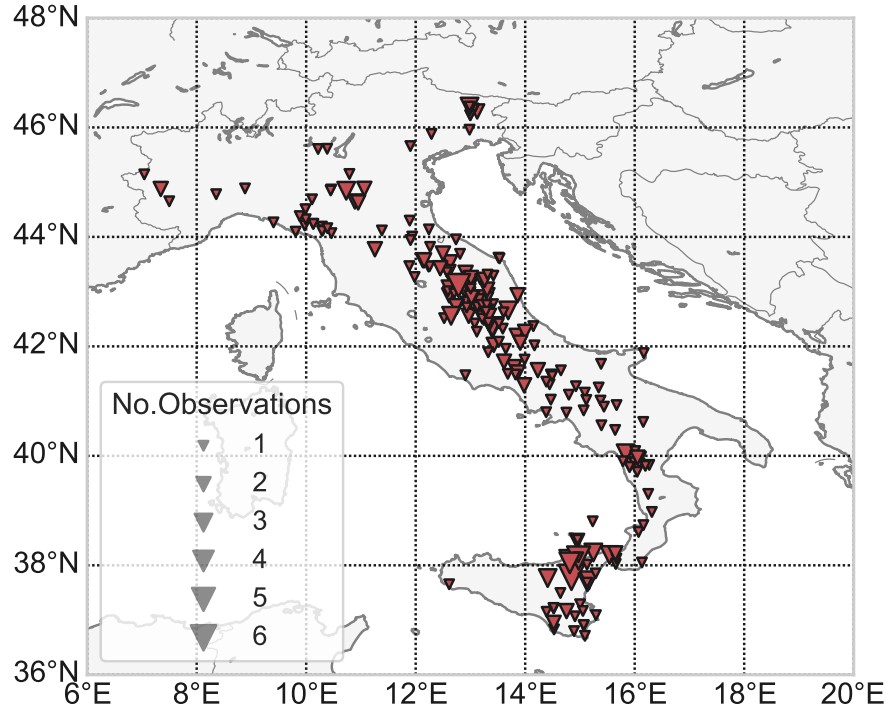

**Figure 4.** Location of the stations associated to the macroseismic data. The size of the symbol is proportional to the number of macroseismic points associated

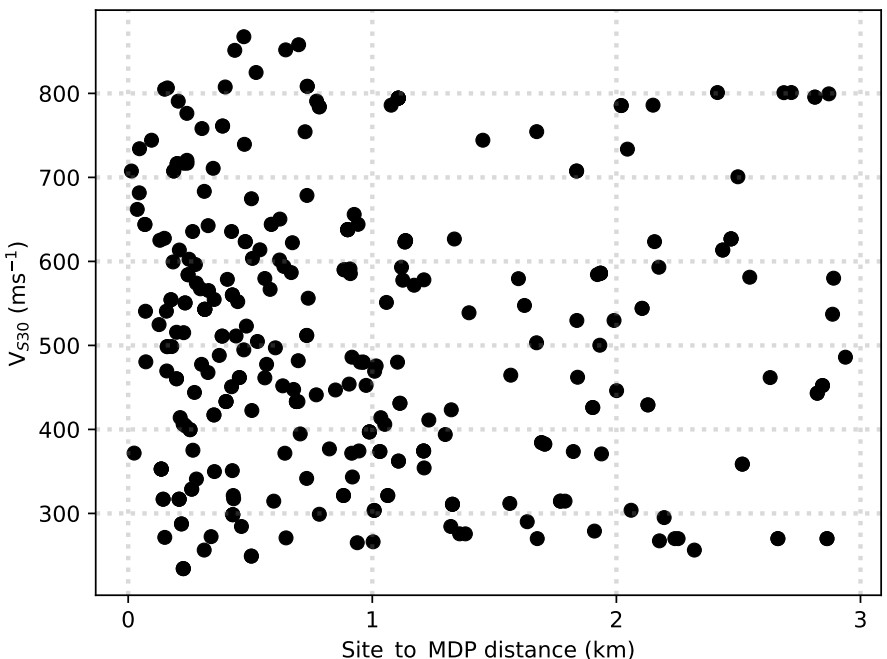

**Figure 5.** $V_{S30}$ of the intensity points and site-to-MDP distance distributions

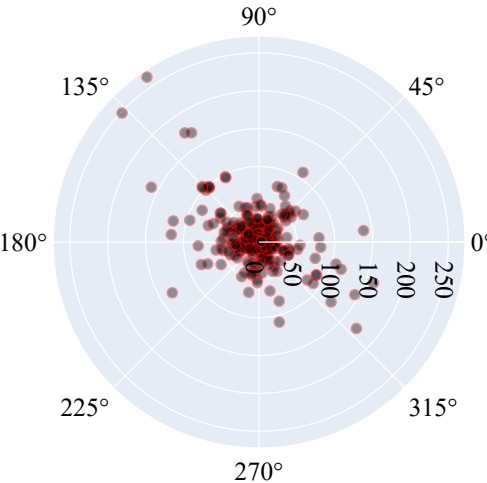

**Figure 6.** Azimuth-distance distribution of the selected stations resulting from the pairing with the macroseismic data points

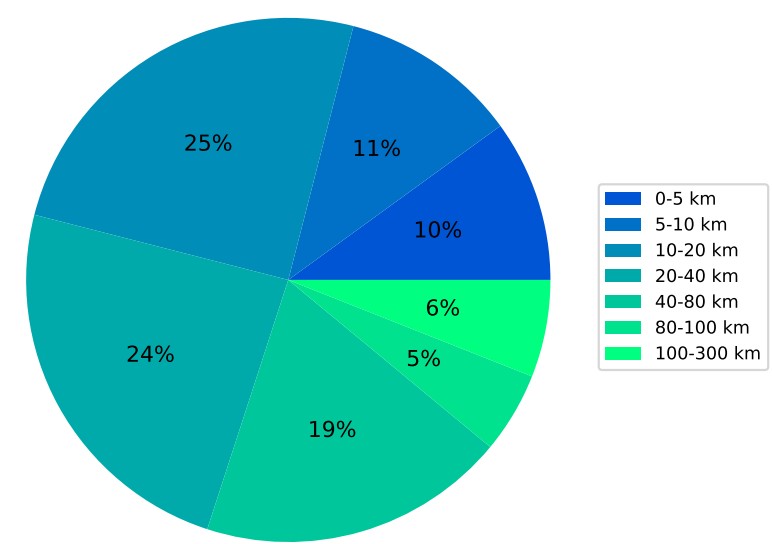

**Figure 7.** Distribution of the source to the recording station distances of the selected data set

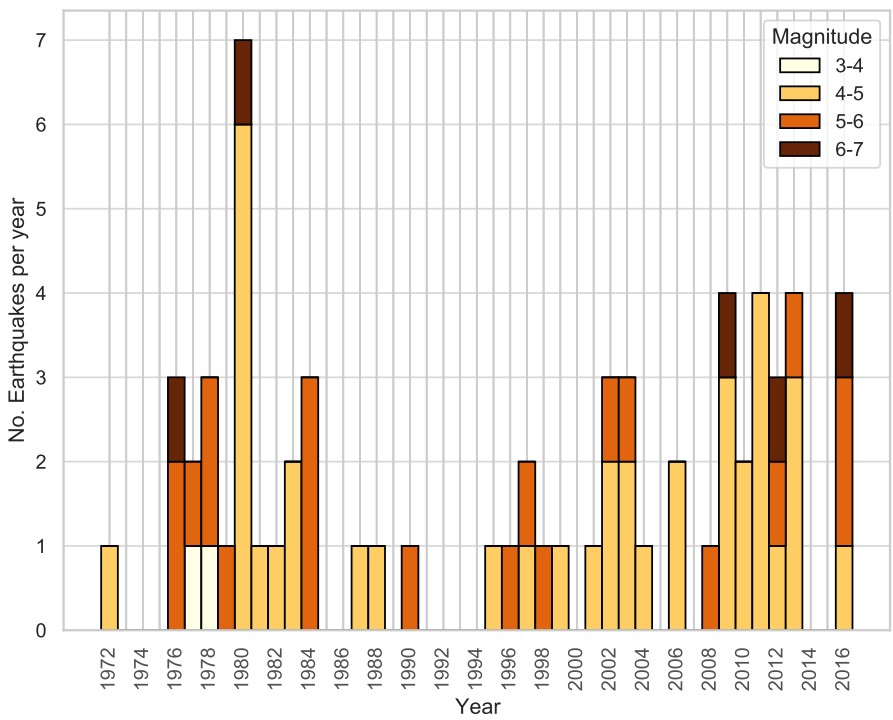

**Figure 8.** Number of earthquakes in the compiled data set in the time interval 1972–2016 for different magnitude ranges



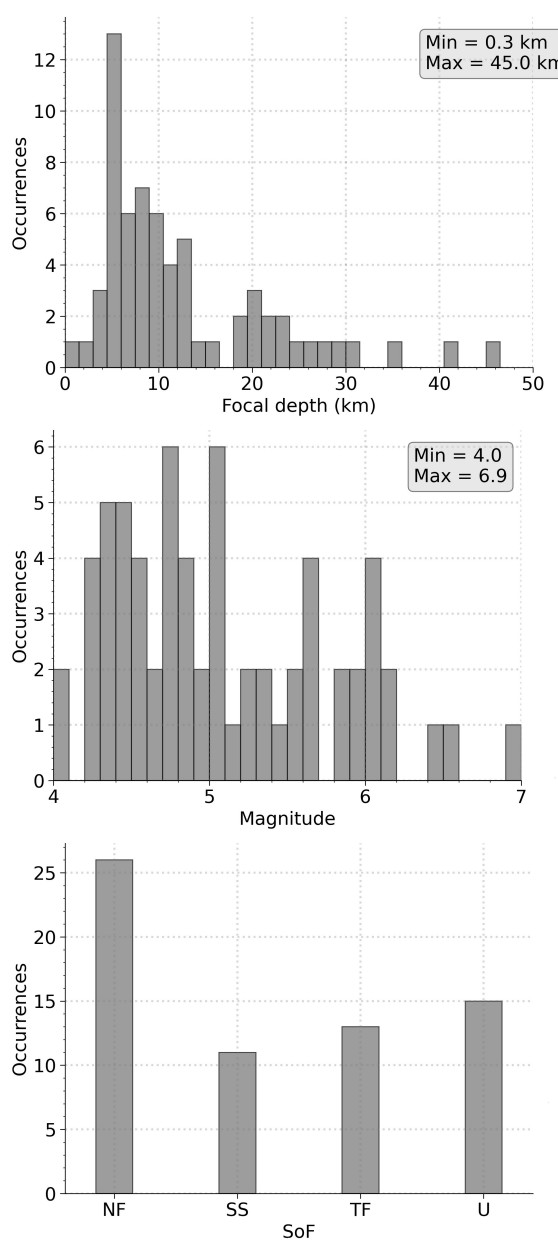

**Figure 9.** Distribution of earthquake (up) depths, (center) magnitudes and (bottom) styles of faulting. U: undefined; SS: strike-slip; TF: thrust; NF: normal faulting




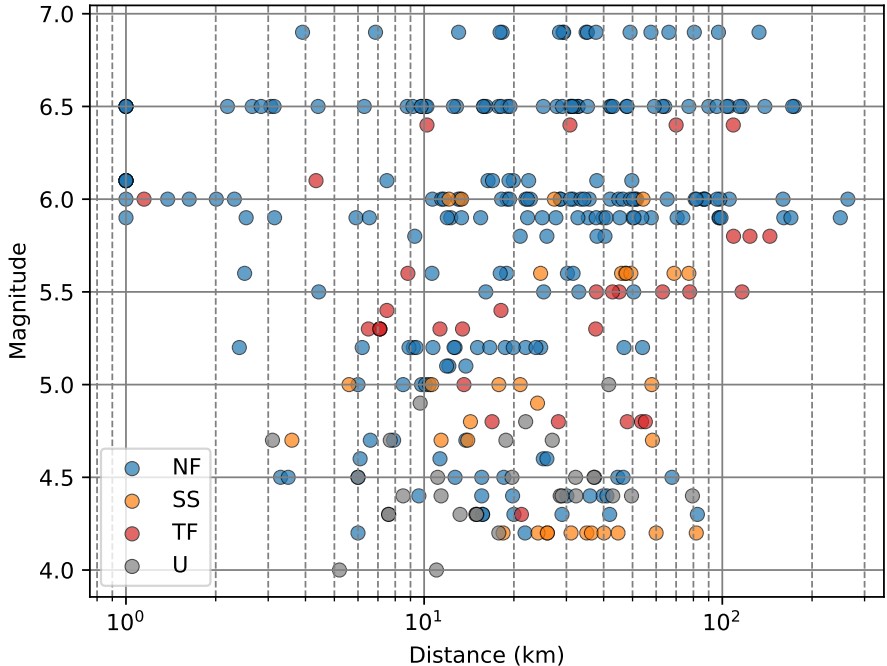

**Figure 10.** Magnitude-distance scatter plot of recordings grouped by style of faulting. U: undefined; SS: strike-slip; TF: thrust; NF: normal faulting. In order to avoid the loss of distance values equal to zero, we assigned a slightly bigger value than zero (1 Km)





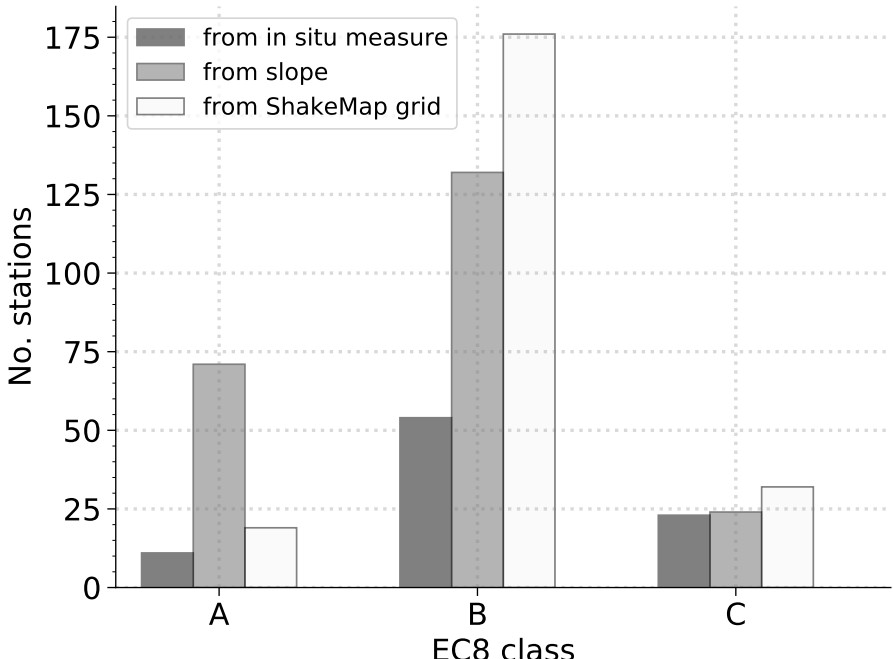

**Figure 11.** Histograms with reference to recording stations showing EC8 site classes



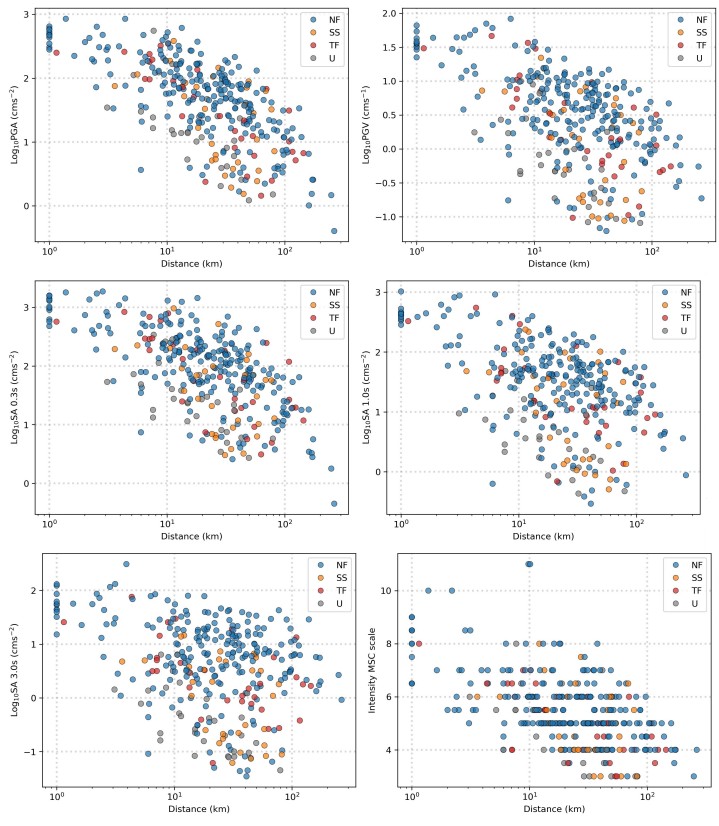

**Figure 12.** The attenuation characteristics of (top left) PGAs, (top right) PGVs, spectral accelerations at periods (center left) 0.3s, (center right) 1.0s and (bottom left) 3.0s, (bottom right) MCS Intensity grouped by style of faulting. U: undefined; SS: strike-slip; TF: thrust; NF: normal faulting. In order to avoid the loss of distance values equal to zero, we assigned a slightly bigger value than zero (1 Km)

*Author contributions.* Oliveti, Faenza, and Michelini wrote the manuscript. Oliveti designed and implemented the dataset and data viewer with input from all of the authors.

*Competing interests.* The authors declare that they have no conflict of interest.

*Acknowledgements.* The INGe dataset includes data obtained from the ESM Database and the DBMI15 Database. The work of the data contributors and the scientific community supporting these databases is gratefully acknowledged.




*Financial support.* This work was funded by the Italian Civil Protection (2019-2021) B2 ShakeMap adjournment project.





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
