# Peer review of "INGe: Intensity-ground motion dataset for Italy"

_Earth System Science Data, 2020_

## Referee Comment (RC1) · Antoine Schlupp (Referee) · 30 Jan 2021

A) General comment.

The benefit or interest doing such a dataset is clear and well explained. Personally, I am convinced as we studied macroseismic and instrumental correlation in the past (Lesueur et al. 2013, ref that could be add as very close to the paper approach, see full reference here under). In that work we faced the difficulty to have numerous seismic stations and macroseismic information separated by short distance to be able to compare them. In the submitted paper and dataset, the very near common observations, less than 3 km and many times less, is great for further studies. Therefore, I support the publication and the built dataset that will save huge time for future users. The paper is well written and explains well the aim of the dataset. I have nevertheless few questions to clarify the text and quite a list of comments or questions about the dataset

itself. They should be checked or clarified before publication, despite the format of the publication is with a short length.

Lesueur, C., M. Cara, O. Scotti, A. Schlupp, and C. Sira. Linking ground motion measurements and macroseismic observations in France: a case study based on accelerometric and macroseismic databases. Journal of Seismology, 17 (2) pp. 313-333, doi 10.1007/s10950-012-9319-2, 2013.

B) Specific comments:

1) There is one main point that must be clarified for a good use of the dataset. The station is placed at a very precise point. But the MDP location corresponds to a "a pair of geographical coordinates matching the average macroseismic intensity value of a more or less large area with a point". How the user will know the extent of the area, is it the whole urban area of the city? If the locality is large, this point could be at more than 3 km despite the station is inside the city. Did you exclude the MDP in that case? This method probably aggregated sometimes areas with a part at rock and a part at "site effect" but the whole zone will be affected to one point. How the user will know if the "location point" calculated is consistent with the whole area? If the calculated point will be at rock when the most area is at sediment, how can we correlate the measurements? Does the VS30 at the MDP corresponds to that specific point or to the average VS30 of the whole area used to determine the intensity? You said at 105-106 that you "check of geological and topographical condition match". Is it for the whole area or for the "reduce" location at a point? As user, I would like really to know that before using the dataset and making conclusions on the results.

2) Line 100: Please clarify in the following sentence what was not reliable, it is not clear for me. " To generate our dataset, we have extracted all the MDPs from DBMI15 corresponding to earthquakes listed in the ESM flatfile and that have not been listed in the Macroseismic Bulletin since this latter data source has been proved to be largely unreliable Âż. You mention also "unreliable" in line 58, but with no explanation. Could

you precise it shortly?

3) The criteria of 3 km is very interesting. But the variation in ground motion with distance (site to MDP distance) is higher at short epicentral distance then long epicentral distance. The criteria could be too large for location very near epicenter (you have very short epicentral distances) and too restrictive for large distance (for example 100 km or more). I understand the need to have one distance value but is seem more important for me to know if the local "site effect" are similar or different between station and MDP. See first specific comment. I advise to include a comment on that point in the text.

4) line 20 to 24: The citations (DYFI of USGS, INGV, EMSC) seems to be related to Italian territory. It should be specified (they are other numerous and high quality "DYFI" procedure in other European countries)

5) line 29: several observatories are already producing shakemaps that include instrumental and macroseismic data (USGS, BCSF at www.franceseisme.fr and many others). They are directly in the topic of the paper bringing together these two types of data. Include at least the information would support the paper, with some citations.

6) line 95 to 99: I did not understand if you are using these data with high uncertainty at the end. Could you clarify it ?

7) line 108: your selection is on the closest one, not on sites conditions. It is your choice well explained here but why not including all of them within the 3 km distance? It would give an idea of variability within this 3km distance.

Comments or question about the dataset:

- the number of decimals in most of the dataset is not homogeneous and sometimes they are 10 and more decimals which has no meaning (for example event latitude at a precision of less than one millimeter? distance between epicenter and MDP much lower than one millimeter? ...). It should be homogenized and with a number of decimal consistent with the uncertainties on the value.

**ESSDD**

- the dataset does not include any uncertainty on the value. This can decrease the use of the dataset because the user will need to correlate this dataset and the original catalog (with the event ID) for instrumental data which is in the opposite of the aim of this work (at least for ev_lat, ev_long, ev_depth, Magnitude)

- the stations are called "site" or "station" or "st" in the header line. It would be better to homogenized to avoid misunderstanding.

- there is a lack of explanation of each column or header meaning. I did not find what is vs30_m_sec_WA for example. It should be available with the dataset.

- the type of distance to epicenter is not explained. Or in other words, are you able to specify when it is Repi and when it is RJB ?

- What does mean the starts behind EC8 code. Not reliable? In that case why?

- a column with "MDP epicentral distance" would have been appreciated even they are not far from instrumental data.

- What is the magnitude type ?

- you have "distance_km" with values of 0 which are not consistent with epicenter location and station location. Is "distance_km" the epicentral distance of the station?

- you have many type of Vs30 values, it would help by specifying which one is measured on the field (I speculate from the value that it is this one "vs30_m_sec")

Some of the requests in this list can be found from original seismic catalog but it is really pity that it is not available directly in this dataset.

C) Technical corrections - line 77 you use "s" and line 78 "Hz" for the "range", I would advise using the same scale.

- Line 94 you do not use Roman number for intensity, and you put one decimal. For Intensity we should always prefer roman number. Same comment for lines 96 and 97.

- line 99: need a ")" after 2007

- line 158, specify the figure number

- line 250: where can we find the 2011 version of DATABASE MACROSISMICO ITAL-IANO, could you add a link ?

- line 271: Could you precise where do we find this version? Is it this version found on web that should be refered as following in that case ? De Rubeis, V., Sbarra, P., Tosi, P., and Sorrentino, D. (2019). Hai Sentito Il Terremoto (HSIT) - Macroseismic intensity database 2007-2018, version 1, https://doi.org/10.13127/HSIT/I.1

- figure 3: variation in magnitude legend not progressive (M4 very small / others)

- Figure 4: I think that "No Observation" means "observation number". You want to say "number of observations" that should be written "Nb. Observations".

- figure 5: for clarification, I would add in the vertical axis MDP => "Vs30 (ms-1) at MDP" or like in the dataset "Vs30_MDP"

- figure 7. The boundaries between proportion is not easy to see with the color used. Add for example a black line in-between. And correlation between color and legend could be impossible for some readers. Either put a label for each "piece of cake" or change color progression.

- figure 10: same as previous comment about epicentral distance. Do you really have 0km distance? With the dataset, it seems not (using event coordinate and station coordinate).

- figure 11: "ShakeMap grid" is from Michelini et al. 2020 ? If yes precise it at least in the figure legend text.

---

## Referee Comment (RC2) · Anonymous Referee #2 · 10 Feb 2021

**Review - MS No.: essd-2020-372**

Title: INGe: Intensity-ground motion dataset for Italy
Author(s): Ilaria Oliveti et al.
MS type: Data description paper

**General Comments**

The publication of an intensity-ground motion dataset is in principle very welcome. It provides a good basis for further research. The dataset also enables better insights into previous studies on empirical relationships between instrumental ground motion data and intensities. The reprint, however, still contains a number of points that require clarification, addition or amendment. Some of these are of a fundamental nature.

**Specific Comments**

**1  line 7**

It would be important to specify the type of magnitude used for the different earthquakes as part of the dataset. Even the given range can be different if you refer to Mw or ML.

**2  lines 21-26**
Different DYFI approaches are discussed and the macroseismic scales they use. The following comments arise in this context:
**2-1  Provide the references for the three scales given in lines 25 and 26.**

**2-2  Use the correct reference for the MM scale used for DYFI.**

**2-3  Regarding the use of the MCS scale in Italy, it would be important for readers outside Italy or for readers who are not exactly insiders to know why the MSC scale, which is about 90 years old, is still used in Italy. If possible, also give the reference to the authorised translation into Italian, as presumably not all users of the scale in the country can use the original.**

**2-4  With regard to the EMS, a distinction must be made between the 1992 and 1998 versions. The EMS-98 version is certainly meant; in this case, the designation must be made accordingly.**

**3  line 54**
What is meant by the slash in the expressions "(PGV)/intensity" and "(SA)/intensity"?

**4  line 57**
It is stated which three spectral acceleration (SA) values are used. On the other hand, chapter 3 deals with practical concerns of the EC8. However, the three used spectral periods are only of limited

relevance with regard to the position of the plateau values of the design spectra of the EC8. Further SA values for periods lower than 0.3 s should be added.

**5 line 57**
It says: ". . we . . .remove several earthquakes whose data were proven to be unreliable." It would be important for readers to know which earthquakes this concerns. The knowledge that the authors have gained in this respect should be shared (e.g. in a table with the related earthquakes).
What are the criteria that resulted in excluding certain earthquakes? Are these criteria or findings that have been expressed before or by other authors? Or is this fundamentally new knowledge?
A certain transparency would be expected.

**6 lines 14-65 (Meaning, this comment refers to the introduction in general.)**
With regard to earlier datasets or works on empirical relations between ground motion data and intensities in Italy, reference is made to Faenza and Michelini (2010, 2011). There are, however, a number of other works specifically for Italy, but also works worth citing for other study areas. Shouldn't these be acknowledged in the introduction?
Further down in the text, Gomez-Capera et al. (2020) are mentioned. They refer to at least nine similar papers, five of them concerning Italy. All these papers are based on data sets obviously similar to the one described here by the authors of the reprint.

**7 lines 92-93**
Here it is stated: ". . . the average macroseismic intensity value of a more or less large area with a point." In the following, individual questions on this:
**7-1 Does "more or less large area" mean the locality to which an MDP applies? Please specify.**

**7-2 On the basis of the data set for the reprint https://doi.org/10.13127/inge.1, the localities are villages, small towns and up to medium-sized towns with their geographical extensions. With respect to Ancona it is the centro storico. Right?**

**7-3 What might be meant by "the average macroseismic intensity value"? What are these average intensity values? And from which values was the average or mean calculated?**
Intensities for a location are determined, as the authors know, from the totality of all macroseismic observations per location - never, as one might think from the wording in the reprint, as the average of point intensities for a location. Only from the totality of all macroseismic observations per location it can be determined in how many cases (or to what percentage) certain effects were observed or not observed. This is because the MCS scale is also based on the use of the frequency (or percentage) of occurrence of certain macroseismic effects per locality, including the types of building damage. The wording in the lines mentioned would therefore have to be corrected or clearly described as to how it was actually done.

**8  line 94 ff.**

**8-1  The intensity is always given in the text, i.e. not only in the line mentioned, with e.g. 6.0, i.e. a decimal number. However, intensities assigned from macroseismic observations are always positive integers from 1 to 12. Therefore, intensities are also indicated with Roman numerals in older literature to emphasise the integer nature of the intensity. However, Roman numerals are somewhat unsuitable for numerical applications. Only derived intensities, i.e. intensities calculated from other quantities by means of empirical relations, appear as decimal numbers.**

Therefore, either change the intensities to integers or justify why decimals are used.

**8-2  It is correct to use the notation 6-7 for cases of uncertain macroseismic findings where the intensity can be determined equally well, e.g., as 6 degrees or 7 degrees. But this notation is only used to express the uncertainty described. It is not a range between 6.0 and 7.0, as it says in the mentioned line. It is also by no means an intensity with the accuracy of half a degree. Otherwise the MCS scale would not have 12 degrees but 23.**

The wording would therefore have to be changed in the reprint.

**9  line 96**

**9-1  Explain the meaning of the abbreviations "(e.g. "HD", "D", or "F")"; i.e. what do "HD" etc. stand for?**

**9-2  What means „class 4.0"? Is "class" used as a synonym for "intensity"? And if so, why?**

**9-3  Aren't classes numbered as natural integers instead of decimals? Please, explain, why you prefer decimals.**

**10  lines 114-118**

**10-1  It would be extremely helpful for users if a table could be added with the 65 earthquakes that are dealt with here. Here all parameters, such as Mw and ML, can be specified (i.e. both magnitudes!), but also the number of available MDP per event (see lines 133-137). Many of these data are contained in the data set, but would have to be extracted first. A special table for the 65 earthquakes, as described, would be very user-friendly. Please add the number of available MDP per event to the table.**

**10-2  The fact that the reprint does not distinguish whether a magnitude is Mw or ML is, as already mentioned, unacceptable. In addition, the Mw values should be available for all quakes considered in the data set. The CPTI15 file (Rovida et al. 2020; BEE vol. 18) lists all earthquakes in Italy with Mw >= 4.0 (either true or proxy). The CPTI15 file also contains the related uncertainty for each Mw value. The uncertainty in magnitudes should also be included in the required table.**

**10-3  Why the geographical coordinates of, e.g., M6 earthquakes are given with an accuracy of up to 7 or 8 digits after the decimal point in view of a considerable extension of the fault plane?**

**11  lines 119-120**

**11-1  Three columns of $v_{S30}$ values are given in the data set: vs30_m_sec, vs30_m_sec_WA, vs30_m_sec_shakemap. These designations would also have to be used in the two rows for clarity as to what is meant. Use a capital S, as it comes from S-wave.**

**11-2  It would be important to add the uncertainties of the $v_{S30}$ values. These differ in part significantly for the three types in the data set. In view of the differences in $v_{S30}$ according to different types of determination at a measuring point of more than 500m/s² in the data set, this information is more or less useless without explicit error information.**
If it is not possible to state the uncertainties for all measurement points and all types explicitly, at least summary estimates for certain data groups should be provided for the users of the data set. For future applications, such uncertainty assessments will play an increasing role.
In this context, the question arises why the $v_{S30}$ in m/s are given in the three columns (vs30_WA, vs30_shakemap, vs30_MDP) with an accuracy of up to 6 or 7 digits after the decimal point (!). How does this supposed accuracy relate to the actual uncertainty of these quantities?

**12  lines 122**

With regard to the distance between the location for which an intensity value is valid and the measuring point of the strong ground motion station (site_to_MDP_distance_km), there can only be rough estimates. An intensity value can only be representative for a locality (up to the size of a medium-sized town), which can have a considerable spatial extension. Is the edge of the town then used to determine the distance or its centre, or what? In any case, the uncertainty of the distance parameter also plays a decisive role here. It is essential to specify this uncertainty.
In the published data set, the distance in km is given with 9 digits after the decimal point (!) - and this with regard to a distance from a measuring point to an area of a town with an extension in the order of a few to several kilometres. Is there a reason for the remarkable accuracy of 9 decimal places?

**13  lines 125-126**

Here it is stated: " . . .  the maximum between the two horizontal components of the peak ground motion measures . . .".
**13-1  What do you mean with "the maximum between the two . . . components"?**
Could it be "the maximum difference between the two components"? Or what?
Or perhaps "the maximum of the two components"?

**13-2  Measurements of PGA, PGV and calculations of derived vales like SA are connected with uncertainties. These have rarely been used so far, but will be of importance in the future. Since the already published data set covers a relatively large time span from 1972 to 2016, at least qualitative information on uncertainties would be unavoidable. It makes a big difference whether data from an early period of strong ground motion measurements are considered (PGA from low dynamic analogue instruments, later somehow digitised) or modern data (PGA measured with 24 bit digitizers). It is obvious that the uncertainty level has changed radically in such a time span.**
This should be taken into account at least as a discussion.

**14  lines 127-129 (macroseismic data)**

**14-1 It is recommended that the already published data set be structured in such a way that, in addition to MCS intensities, those according to EMS-98 can also be included. On the one hand, this would ensure an opening with regard to other parts/countries of Europe where the EMS-98 is routinely used. On the other hand, such an extension could take into account the current developments in Italy to increasingly use the EMS-98 in macroseismology. This is expressed in the fact that more recent earthquakes are increasingly being processed using the EMS-98. Previous earthquakes, such as the 1976 Friuli earthquake of 6 May 1976, have been re-evaluated using the EMS-98 (Tertulliani et al. 2018; BGTA). Recent ground motion-to-intensity conversion equations (i.e., the application domain of the reprint reviewed here including dataset) also use intensity data in Italy in terms of EMS-98 (Zanini et al. 2019; Eng. Struct.) with data from 1983-2016.**

It is true that, according to Musson et al. (2010), intensity assessments according to MSC and EMS-98 are comparable in principle, but in detail and regarding concrete MDPs, some differences become clearly visible ( cf. Tertulliani et al. 2018). So, there would actually be no reason to ignore the described development of also using the EMS-98.

**14-2  $v_{S30}$ should, according to the original definition, be a point information, i.e. the borehole location at which $v_{S30}$ was determined. If, on the other hand, a single value is given for $v_{S30}$ that is representative for the area of a medium-sized town, the error range of this information must not be neglected under any circumstances. This is because in many cases a considerable areal variation of this parameter is observed in the region of such a town. In the data set, $v_{S30}$ is nevertheless also given here with an accuracy of 6 digits after the decimal point.**

**15  line 133**

Fig. 1 (p. 7) is mentioned here. In the figure, the assigned intensities are given for 3-4 to 10-11. These should therefore be intensity specifications, as it says in line 92, for the cases in which a clear intensity assignment for one degree of intensity in the form of an integer value is not possible. Why are only these uncertain indications shown graphically and not the intensities of the integer values from 4 to 10? These intensities are also more frequent in the data set than the uncertain ones with e.g. 6-7. For reasons of practicability, these can be shown with e.g. 6.5, but it should be clear what we are dealing with such a notation: an uncertain indication as a proxy for 6-7 but in no case an intensity determined exactly to half a degree.

Fig. 1 should be changed accordingly.

**16  line 135**

Here is referred to Fig. 2. The same applies here as for Fig. 1 (cf. #15).

**17  line 177**

It is said: „ . . . small intensity values (i.e. in the range 3 ≤ MCS ≤ 3.5)". Here again the at least implicit use of the integer values of intensity as a decimal quantity occurs. There exist in that case only the integer values of 3 and 4. If the observational data are so poor that it can be both 3 or 4 degrees, one writes 3-4. But what does the MCS scale provide for any values in between?

**18  line 186**
A data set with magnitudes of 4.2-6.9 is mentioned here. In the abstract, the data set starts at 4.0. Which is correct?

**19  line 187-188**
It is said: „ The dataset can be used as reference to benchmark studies seeking correlations between ground motion parameters and MCS macroseismic intensities". That is correct. However, the reader would have liked to see graphs showing exactly these data points. Preferably supplemented by the empirical adjustments based on earlier studies from Faenza & Michelini (2010) to Gomez-Capera et al. (2020); i.e. the graphical representations of the derived empirical relations. This would give a first, albeit only visual, impression of how the new data set behaves. Such a supplement would be very useful.

**20  line 198-199**
The use of the data for the determination of Intensity Prediction Equations, as it is called here, should hardly be possible without precisely defined magnitudes in the data set, i.e. not knowing what type of magnitude we are dealing with in individual cases. Compare earlier comments on the specification of magnitude types.

**Technical Corrections**

**21  line 67**

"ESM" needs also to be explained in the main body of the text, not only on the abstract.

**22   line 79 ff.**
Mean shear wave velocity of the uppermost 30 m is given as "$V_{S30}$":
However, the derived physical quantity of velocity is abbreviated with a small v according to ISO (International Organization for Standardization). Unlike PGV, the small v has therefore be used for $v_{S30}$. A change according to the ISO standard is recommended.

**23   line 79 and other occurrences below**
With regard to Eurocode 8, it is better not to quote "Code (2005)" but "Eurocode 8 (2005)". Further below, EC8 is used; however, without explanation of the abbreviation.

**24  line 158**
Here it is given "Fig ??b". Insert the appropriate number.

---

## Author Comment (AC1) · 20 Mar 2021

Authors' response to Referees' comments

Dear referees, we would like to thank you for the time and effort put into reviewing our work and we appreciate the constructive criticism that allowed us to make improvements. This is precisely the level of scrutiny we hoped for. We have modified the manuscript and the data set according to your comments, corrections and suggestions. Our response (marked with R: and formatted in blue text) addresses thoroughly all the comments. The updated data set can be downloaded from data repository Zenodo at <a href="https://doi.org/10.13127/inge.2">https://doi.org/10.13127/inge.2</a>

Best regards,

Ilaria Oliveti on behalf of all co-authors

**Referee 1**

A) General comment.

The benefit or interest doing such a dataset is clear and well explained. Personally, I am convinced as we studied macroseismic and instrumental correlation in the past (Lesueur et al. 2013, ref that could be add as very close to the paper approach, see full reference here under). In that work we faced the difficulty to have numerous seismic stations and macroseismic information separated by short distance to be able to compare them. In the submitted paper and dataset, the very near common observations, less than 3 km and many times less, is great for further studies. Therefore, I support the publication and the built dataset that will save huge time for future users. The paper is well written and explains well the aim of the dataset. I have nevertheless few questions to clarify the text and quite a list of comments or questions about the dataset.

Lesueur, C., M. Cara, O. Scotti, A. Schlupp, and C. Sira. Linking ground motion measurements and macroseismic observations in France: a case study based on accelerometric and macroseismic databases. Journal of Seismology, 17 (2) pp. 313-333, doi 10.1007/s10950-012-9319-2, 2013.

R: Thank you very much for this positive feedback. We have added the citation in the introduction (page 2, line 32).

**B) Specific comments:**

1) There is one main point that must be clarified for a good use of the dataset. The station is placed at a very precise point. But the MDP location corresponds to a "a pair of geographical coordinates matching the average macroseismic intensity value of a more or less large area with a point". How the user will know the extent of the area, is it the whole urban area of the city? If the locality is large, this point could be at more than 3 km despite the station is inside the city. Did you exclude the MDP in that case?

This method probably aggregated sometimes areas with a part at rock and a part at "site effect" but the whole zone will be affected to one point. How the user will know if the "location point" calculated is consistent with the whole area? If the calculated point will be at rock when the most area is at sediment, how can we correlate the measurements? Does the VS30 at the MDP corresponds to that specific point or to the average VS30 of the whole area used to determine the intensity? You said at 105-106 that you "check of geological and topographical condition match". Is it for the whole area or for the "reduce" location at a point? As user, I would like really to know that before using the dataset and making conclusions on the results.

R: Thank you very much for pointing this out. We are conscious of the limitation inherent to this kind of matching.

We have added the following comment to the paper to clarify this point (page 4, line 108):

"The main issue relates to the intrinsic high spatial variability of the two different types of ground shaking values: the instrumental recording provides a geographical point estimation of the ground motion and the measurements depend on the local site conditions; on the other hand, the several intensity observations that contribute to assigning a unique intensity level to a locality are taken on an extended urbanized area which may have different geological, geomorphological, and topographic characteristics. Thus, unlike ground-motion measurements, they do not exist at a point (Worden et al., 2010) because intensity is a classification of the severity of the effects caused by the ground shaking on a "statistically" consistent sample of buildings inside the locality. We took into account this aspect by introducing an uncertainty of 0.5 to all the intensity values".

(see full reference here: Worden, C. B., D. J.Wald, T. I. Allen, K. Lin, D. Garcia, and G. Cua (2010). A revised ground-motion and intensity interpolation scheme for ShakeMap, Bull. Seismol. Soc. Am. 100, no. 6, 3083–3096.)

Unfortunately, DBMI15 does not provide the extent of each locality and advises users to search for a specific locality by the administrative subdivision where it is today located, as it is officially reported by the National Institute of Statistics in the year 2015. In this way, it is possible to know how many localities belong to the same ISTAT code (e.g., 92 localities are associated with Rome). Although important, this information does not allow us to extrapolate each locality's actual size correctly and does not help us improve the matching in terms of local effects because, as explained above, intensity is a classification of the severity of the effects caused by the ground shaking on a "statistically" consistent sample of buildings whose actual location inside the locality is unknown.

In addition, it is worth noting that since each MDP results from several intensity estimations within the same urbanized area, it would make little sense and it could be potentially misleading to adopt the  $v_{s30}$  extracted at the MDP. For this reason, we have decided to remove this punctual value from the data set because it would likely lead to confusion to the users adopting the dataset.

With regard to the sentence "check of geological and topographical condition match", we referred to the work of Gomez-Capera et al. (2020) in which the association is carried out on the basis of the same distance criterion but, unlike our work, they prefer, within the threshold of 3 km, the "locality point" with similar topographic conditions respect to the station.

We have revised this to read:

"we have chosen a distance criterion, common to most of the studies in this field (e.g., Caprio et al., 2015; Locati et al., 2017) while, recently, Gomez-Capera et al. (2020) correlated these observations keeping a threshold distance of about 3 km and, at the same time, preferring the locality with similar topographic conditions respect to the station"

In summary, based on the partial information provided by DBMI15, we have chosen the distance criterion, common to most of the studies in this field; to compensate for the limitations listed above, we have assigned an uncertainty of  $\pm 0.5$  to the intensity values.

In our dataset, however, we opted to report all the MDPs within 3 km from the station leaving to the user to make the final selection (see below).

2) Line 100: Please clarify in the following sentence what was not reliable, it is not clear for me. " To generate our dataset, we have extracted all the MDPs from DBMI15 corresponding to earthquakes listed in the ESM flatfile and that have not been listed in the Macroseismic Bulletin since this latter data source has been proved to be largely unreliable". You mention also "unreliable" in line 58, but with no explanation. Could you precise it shortly?

R: We have included the following additional information:

"...listed in the ESM flatfile and that have not been listed in the Macroseismic Bulletin. This latter data source has been proved to be largely unreliable because they have been provided by non-practitioners (e.g., staff personnel of the public administration in Italy) in the estimation of the macroseismic intensities.

More specifically about the Macroseismic Bulletin, INGV since long manages, in the post-earthquake, a correspondent network to record the earthquake's effects on people, buildings, and the environment. The network supplies information related to their environment through the compilation of questionnaires based on

MCS scale. The network includes public entities such as the municipal staff, the forestry corp and the carabineers corp who are not macroseismic assessment practitioners.

3) The criteria of 3 km is very interesting. But the variation in ground motion with distance (site to MDP distance) is higher at short epicentral distance then long epicentral distance. The criteria could be too large for location very near epicenter (you have very short epicentral distances) and too restrictive for large distance (for example 100 km or more). I understand the need to have one distance value but is seem more important for me to know if the local "site effect" are similar or different between station and MDP. See first specific comment. I advise to include a comment on that point in the text.

R: We have addressed the issue of the choice of the distance criterion in the response to the first comment in order to clarify that the available information does not allow us to take site effects into account in a reliable way.

Regarding the variation in ground motion with distance, it can be seen by plotting the station-to-macroseismic data point distance versus the station-to-event distance (see the new added Figure 9). It is possible to notice that, selecting only the closest intensity points for each recording station (indicated by bigger solid circles), the distance between macroseismic observations and ground-motion stations tends to grow with the Joyner-Boore/epicentral distance, to show that this latter distance criterion can be suitable for our data.

We have added the following comment into the manuscript to clarify this issue and to describe the new added Figure 9:

"The distribution of all the MDPs in the station-to-macroseismic data point distance and the stationto-event distance is illustrated in Fig. 9. The station-to-MDP distances range between 0.01 km and 2.99 km. This figure evidences that most intensity-PGM pairs, selecting only the closest intensity points for each recording station (indicated by bigger dots), do not distance more than 1.5 km and are rather homogeneously distributed in terms of Mw. Moreover, it is important to note that the distance between macroseismic observations and ground-motion stations is lower for pairs with short station-to-event distance and higher for those having large station-to-event distance. This all evidences that the distance criterion of the 3 km appears suitable for our data."

4) line 20 to 24: The citations (DYFI of USGS, INGV, EMSC) seems to be related to Italian territory. It should be specified (they are other numerous and high quality "DYFI" procedure in other European countries)

R: We have updated the proposed citations to provide a more global overview on this issue. The modified sentence is now:

"...[e.g., internet-based questionnaires, such as the U.S. Geological Survey (USGS) "Did You Feel It?" (DYFI) system (Quitoriano and Wald 2020) and the Italian "Hai Sentito Il Terremoto?" HSIT database (Tosi et al., 2007); the LastQuake smartphone app developed by the European Mediterranean Seismological Centre (EMSC) for global earthquake eyewitnesses (Bossu et al., 2017)]"

The related full references are listed here below:

Quitoriano, V., and Wald, D. J. (2020) USGS "Did you feel it?"-science and lessons from twenty years of citizien science-based macroseismology. Front. Earth Sci. 8:120.

Bossu, R., Landès, M., Roussel, F., Steed, R., Mazet-Roux, G., Martin, S. S., et al. (2017). Thumbnail- based questionnaires for the rapid and efficient collection of Macroseismic Data from Global Earthquakes. Seism. Res. Lett. 88, 72–81.

5) line 29: several observatories are already producing shakemaps that include instrumental and macroseismic data (USGS, BCSF at www.franceseisme.fr and many others). They are directly in the topic of the paper bringing together these two types of data. Include at least the information would support the paper, with some citations.

R: We have added the following additional information into the manuscript (page 2, line 29)

"To this purpose, the ShakeMap software (Wald et al., 1999), developed by the U.S. Geological Survey (USGS), was adopted by several operational centers worldwide (e.g., in Europe at National Institute for Earth Physics, Romania, Sokolov et al., 2009; Institute of Engineering Seismology and Earthquake Engineering [ITSAK], Greece, Theodoulidis et al., 2019; Bureau Central Sismologique Français - Réseau National de Surveillance Sismique, France, Schlupp and Grunberg, 2018)".

The related full references are listed here below:

Wald, D. J., V. Quitoriano, T. H. Heaton, H. Kanamori, C. W. Scrivner, and C. B. Worden (1999). Trinet 'ShakeMaps': Rapid generation of peak ground motion and intensity maps for earthquakes in southern California, Earthq. Spectra 15, 537.

Sokolov, V. Y., F. Wenzel, and R. Mohindra (2009). Probabilistic seismic hazard assessment for Romania and sensitivity analysis: A case of joint consideration of intermediate-depth (Vrancea) and shallow (crustal) seismicity, Soil. Dynam. Earthq. Eng. 29, 364–381

Theodoulidis, N., K. Morfidis, K. Konstantinidou, B. Margaris, and Ch. Papaioannou (2019). ShakeMaps and rapid earthquake damage assessment in Greece, Proc. of the 2nd International Conference on Natural Hazards & Infrastructure, Chania, Greece, 23–26 June 2019.

Schlupp, A., and M. Grunberg (2018). ShakeMap based on instrumental and macroseismic data in France: Feedbacks on modified V3.5 and expectation on V4, 2018 Seismology of the Americas Meeting, Latin American and Caribbean Seismological Commission Seismological Society of America, Miami, Florida, 14–17 May 2018.

6) line 95 to 99: I did not understand if you are using these data with high uncertainty at the end. Could you clarify it?

R: We have also included these data in our data set. In order to allow the user to recognize them, we have kept the available non-conventional descriptive values (e.g. "HD", "D", or "F") in the column headed "Int". A strategy on how to use these unconventional intensities can be found in the manual describing the DBMI15 (https://emidius.mi.ingv.it/CPTI15-DBMI15/description\_DBMI15\_en.htm).

7) line 108: your selection is on the closest one, not on sites conditions. It is your choice well explained here but why not including all of them within the 3 km distance? It would give an idea of variability within this 3km distance.

R: Thank you very much for this interesting suggestion. We agree and we have included all the localities reporting intensity data which are located within 3 km from the strong motion stations. The new PGM/intensity data set was assembled in this way, collecting a total of 519 observations data pairs. Thus, we have updated most of the figures (Figs.1,2,4,5,6,9) in order to show all the MDPs, allowing, in some cases, also users to distinguish the closest ones with respect to the same recording stations.

The updated figures are listed in the last part of the document.

Comments or question about the dataset:

8) the number of decimals in most of the dataset is not homogeneous and sometimes they are 10 and more decimals which has no meaning (for example event latitude at a precision of less than one millimeter? distance between epicenter and MDP much lower than one millimeter? ...). It should be homogenized and with a number of decimal consistent with the uncertainties on the value.

R: Thank you for pointing this out. This has been fixed. Unfortunately, it derived from the export of the original python pandas dataframe table that did not visualized all the additional decimals which were instead included into the resulting 'csv' table.

9) the dataset does not include any uncertainty on the value. This can decrease the use of the dataset because the user will need to correlate this dataset and the original catalog (with the event ID) for instrumental data which is in the opposite of the aim of this work (at least for ev\_lat, ev\_long, ev\_depth, Magnitude).

R: Thank you for pointing this out. We agree and we have included the uncertainty on the values of the mentioned parameters. In order to supplement the incomplete event information extracted from the ESM flatfile, we based our gathering of the data uncertainty on other sources.

In summary, when assembling our data set, we used the ESM database as the primary source of the parametric earthquake information. This database, however, does not report uncertainties but it only cites the source of the information. When we searched for the uncertainties in the cited sources, we noticed that not all were reported. This occurred especially for the "older" earthquakes. This all led us to change our strategy and we started looking for all those sources that also reported the uncertainties associated with their locations, obliging us to select (in some cases) sources and associated parameters different from those adopted in the first version of the INGe data set. The consequence of this change is that the epicentral distance may have changed (not the Rjb one). We also faced a similar problem for the magnitude. To solve it, we used a single homogeneous catalog in Mw, HORUS (http://horus.bo.ingv.it/). Also, in this case, the magnitudes differ from the first version of our data set. In the new version of the manuscript, all figures have been redone according to the new values. All the updated event metadata (the magnitude and the event location) are fully referenced, introducing two specific fields named "ev\_hyp\_ref" and "Mw\_ref", respectively, for each estimate to allow the traceability of the information source.

10) the stations are called "site" or "station" or "st" in the header line. It would be better to homogenized to avoid misunderstanding.

**R: "site" and "station" have been changed to "st".**

11) there is a lack of explanation of each column or header meaning. I did not find what is vs30\_m\_sec\_WA for example. It should be available with the dataset.

R: Once uploaded the revised data set to Zenodo, we have included the detailed legend presented below in the section dedicated to the description of the dataset. Furthermore, it is worth stressing that we have removed the column headed vs30\_m\_sec\_WA containing  $v_{s30}$  values at the station inferred from the slope, according to Wald and Allen (2007) in m/s. The other values of  $v_{S30}$  have been combined into a single column. An additional column ("st vs30 type") provides a description of how the corresponding values have been obtained. The V830 values are either measured ones as provided by the ESM DB (https://esm.mi.ingv.it//esmws/shakemap/1/masterstationlist.txt) or obtained from the vs30 grid used by the software ShakeMap and described by Michelini et al. (2020) and available on http://shakemap.ingv.it.

**LEGEND CAPTION:**

- **event\_id**: id of the event in ESM (http://esm.mi.ingv.it);
- **INGV\_ev\_id**: event id in the bulletin of the Istituto Nazionale di Geofisica e Vulcanologia (INGV, http://cnt.rm.ingv.it);
- event\_time: time of the event (format YYYY-MM-DD HH:MM:SS);
- ev\_latitude and ev\_longitude: geographic coordinates (decimal degrees) of the epicenter of the event;
- **ev\_depth**: depth of the hypocenter of the event (km);
- **erh**: horizontal error (km);
- **erz**: vertical error (km);
- ev\_hyp\_ref: reference for ev\_latitude, ev\_longitude and ev\_depth (CPTI https://emidius.mi.ingv.it/CPTI15-DBMI15/; INGV\_W INGV-webservice http://webservices.rm.ingv.it; CSI Catalogo della Sismicità Italiana https://csi.rm.ingv.it/; GEM ISC-

GEM Global Instrumental Earthquake Catalogue http://www.isc.ac.uk/iscgem/; ETNA1 Mt. Etna Seismic Catalog 2000-2010 http://sismoweb.ct.ingv.it/Etna/catalogs/2000\_2010/; ETNA 2 Mt. Etna Seismic Catalog 2011-2013 http://sismoweb.ct.ingv.it/Etna/catalogs/2011\_2013/; SL Slejko, D., Neri, G., Orozova, I., Renner, G., & Wyss, M. (1999). Stress Field in Friuli (NE Italy) from Fault Plane Solutions of Activity Following the 1976 Main Shock. Bulletin of the Seismological Society of America, 89(4), 1037–1052. BSING: Istituto Nazionale di Geofisica. Bollettino sismico mensile.)

- **fm\_type\_code**: style of faulting (NF normal fault; TF reverse fault; SS strike-slip fault; O oblique fault; U undefined).
- **Mw**: moment magnitude;
- **erMw**: uncertainty of **Mw**;
- **Mw\_ref**: reference for **Mw** estimate (H: The Homogenized Instrumental Seismic Catalog (HORUS));
- **network\_code**: code associated with the recording network according to the International Federation of Seismograph Network (http://www.fdsn.org);
- **st\_code**: 3 to 5 characters associated to the station;
- **st\_latitude** and **st\_longitude**: geographic coordinates (decimal degrees) of the station;
- **ec8\_code**: EC8 site category (CEN, 2003) (the subsoil class values are marked with an asterisk when they were assessed only by the surface geology)
- **st\_vs30**: vS30 associated to the station (m/s);
- **st\_vs30\_type**: type of method used to estimate **st\_vs30** (meas: average shear wave velocity vS,30 (CEN, 2003) from in-situ measurements (m/s); calc: extracted vS30 from the vS30 grid adopted by ShakeMap (Michelini et al., 2020);
- **st\_to\_ev\_dist**: station-to-event distance (km);
- **st\_to\_ev\_dist\_ type**: type of **st\_to\_ev\_dist** (RJB: distance computed from the surface projection of the fault; REPI: distance from epicentre of the event);
- **epi\_az**: event-to-station azimuth (degrees);
- **PlaceID**: id of the locality to which each macroseismic data point is referring to;
- **Placename**: place name of the locality to which each macroseismic data point is referring to;
- Lat and Lon: geographic coordinates (decimal degrees) of the macroseismic data point;
- Int: macroseismic intensity associated to the macroseismic data point;
- **Int\_dec**: corresponding decimal value of **Int**, used internally for graphing, and based on the DBMI15 manual (https://emidius.mi.ingv.it/CPTI15-DBMI15/description\_DBMI15\_en.htm);
- **erInt**: uncertainty of **Int**;
- **Int\_type**: type of macroseismic scale used to estimate **Int** (MCS: Mercalli-Cancani-Sieberg scale (Sieberg, 1930); EMS-98: European Macroseismic Scale (Grünthal, 1998));
- **st\_to\_MDP\_dist**: station-to-macroseismic data point distance (km);
- ev\_to\_MDP\_dist: epicenter-to- macroseismic data point distance (km);
- Max\_PGA and Max\_PGV: maximum among the two horizontal components of peak ground acceleration (cm/s2) and peak ground velocity (cm/s);
- Max\_TO\_300, Max\_T1\_000 and Max\_T3\_000: maximum among the two horizontal components of peak response spectral acceleration amplitudes at 0.3, 1.0, and 3.0 s (cm/s2);

12) the type of distance to epicenter is not explained. Or in other words, are you able to specify when it is Repi and when it is RJB?

R: We have introduced a specific field named "st\_to\_ev\_dist\_type" in the csv file to do this.

13) What does mean the starts behind EC8 code. Not reliable? In that case why?

R: As explained in the added legend, the subsoil class values are marked with an asterisk when they were assessed only by the surface geology.

14) a column with "MDP epicentral distance" would have been appreciated even they are not far from instrumental data.

R: We have included the column headed "ev to MDP dist" in the csv file as advised.

15) What is the magnitude type?

R: In order to show the uncertainty for all the magnitudes, we have decided to adopt the moment magnitude for each earthquake. In the revised database, the magnitudes are homogeneous, have an associated uncertainty, and come from the same catalog, HORUS catalogue (see previous reply to comment 9 and reference below).

New Reference: Lolli B., Randazzo D., Vannucci G., Gasperini P. (2020). The Homogenized Instrumental Seismic Catalog (HORUS) of Italy from 1960 to Present, Seismol. Res. Lett, doi: 10.1785/0220200148.

16) you have "distance\_km" with values of 0 which are not consistent with epicenter location and station location. Is "distance km" the epicentral distance of the station?

R: We referred to a generic "distance\_km" matching either RJB or REPI according to the following described procedure: when the Joyner-Boore distance is not available, the epicentral distance is used. As advised, we have introduced a field to specify when either RJB or REPI is selected. "distance\_km" with values of 0 was referred to RJB. This is the case for stations that fall within the surface projection of the fault. The column heading name "distance\_km" has been replaced by "st\_to\_ev\_dist", as explained in the legend above.

17) you have many type of Vs30 values, it would help by specifying which one is measured on the field (I speculate from the value that it is this one "vs30\_m\_sec")

R: As explained above, we have included a detailed legend in the section dedicated to the description of the dataset on Zenodo. In addition, we have decided to provide only in situ at the station measured values or values extracted from the  $v_{S30}$  map adopted by new configuration of shakemap (Michelini et al. 2020 and see previousreply to comment 11)

Some of the requests in this list can be found from original seismic catalog but it is really pity that it is not available directly in this dataset.

The information pointed out by the reviewer has been now inserted in the dataset.

**C) Technical corrections**

18) line 77 you use "s" and line 78 "Hz" for the "range", I would advise using the same scale.

**R: We have revised this to read:**

"The periods at which the spectral amplitudes of the (5% damping) acceleration and displacement response are computed range 0.01-10 s, whereas the amplitudes of the Fourier spectrum range 0.02–25 s".

19) Line 94 you do not use Roman number for intensity, and you put one decimal. For Intensity we should always prefer roman number. Same comment for lines 96 and 97.

R: Thank you for pointing this out. We have decided to adopt the standard used in DBMI15 and proposed by AHEAD for intensities, which foresees the use of Arabic numbers (e.g., 8, 9). Furthermore, when the available information is not considered sufficient for assessing an intensity, DBMI15 adopts a dash for expressing uncertain degrees (e.g., 6-7) or non-conventional descriptive codes such as "D" for damage, or "F" for felt. In

these latter cases, for reasons of practicability, taking the approach used in DBMI15 as example, the corresponding decimal value has been reported in an added column headed "Int\_dec".

To address the reviewers' concern regarding clarity about the use of the decimals, in the revised manuscript, we now write:

"For parametrization purposes, macroseismic data with intensity expressed as non-numerical codes ("HF" for Highly Felt, "SD" for Slightly Damage, "D" for Damage, "HD" for Heavy Damage) were converted to numerical values as in DBMI15 and described in Locati et al. (2019, 2021), as reported in https://emidius.mi.ingv.it/CPTI15-DBMI15/description\_DBMI15\_en.htm. According to this approach, the converted numerical value was rounded to the closest half degree (F = 4.0, HF = 5.0, SD = 5.5, D = 6.5, HD = 8.5). For reasons of practicability, also when the available information is not detailed enough to assess an intensity degree in a straightforward way, and such an uncertainty is expressed with a range (e.g., 6-7, 7-8), we give it as decimals (e.g., 6.5, 7.5)"

The related full references are listed here below:

Locati M., Camassi R., Rovida A., Ercolani E., Bernardini F., Castelli V., Caracciolo C.H., Tertulliani A., Rossi A., Azzaro R., D'Amico S., Antonucci A., 2019. Database Macrosismico Italiano (DBMI15), versione 2.0. Istituto Nazionale di Geofisica e Vulcanologia (INGV). https://doi.org/10.13127/DBMI/DBMI15.2

Locati M., Camassi R., Rovida A., Ercolani E., Bernardini F., Castelli V., Caracciolo C.H., Tertulliani A., Rossi A., Azzaro R., D'Amico S., Antonucci A., 2021. Database Macrosismico Italiano (DBMI15), versione 3.0. Istituto Nazionale di Geofisica e Vulcanologia (INGV). https://doi.org/10.13127/DBMI/DBMI15.3

20) line 99: need a ")" after 2007

R: We thank the reviewer for catching the typo. This has been fixed.

21) line 158, specify the figure number

R: We thank the reviewer for catching the typo. This has been fixed.

22) line 250: where can we find the 2011 version of DATABASE MACROSISMICO ITALIANO, could you add a link?

R: As advised, we have included the corresponding link (http://emidius.mi.ingv.it/DBMI11) at the end of the reference.

23) line 271: Could you precise where do we find this version? Is it this version found on web that should be refered as following in that case? De Rubeis, V., Sbarra, P., Tosi, P., and Sorrentino, D. (2019). Hai Sentito II Terremoto (HSIT) - Macroseismic intensity database 2007-2018, version 1, https://doi.org/10.13127/HSIT/I.1

R: The reference has been updated to "Tosi, P., De Rubeis, V., Sbarra, P., and Sorrentino, D.: Hai Sentito II Terremoto (HSIT). Istituto Nazionale di Geofisica e Vulcanologia (INGV). 270 https://doi.org/10.13127/HSIT, 2007."

24) figure 3: variation in magnitude legend not progressive (M4 very small / others)

R: We have updated Figure 3 to modify the point size legend.

25) Figure 4: I think that "No Observation" means "observation number". You want to say "number of observations" that should be written "Nb. Observations".

R: As suggested, "No Observation" has been changed to "Nb. Observations".

26) figure 5: for clarification, I would add in the vertical axis MDP => "Vs30 (ms-1) at MDP" or like in the dataset "Vs30 MDP"

R: We have removed this figure because we have decided to not include  $v_{S30}$  values associated to the macroseismic observations as explained above. Please see our response to specific comment 1.

27) figure 7. The boundaries between proportion is not easy to see with the color used. Add for example a black line in-between. And correlation between color and legend could be impossible for some readers. Either put a label for each "piece of cake" or change color progression.

R: It has been corrected in new Figure 6 of the revised manuscript.

28) figure 10: same as previous comment about epicentral distance. Do you really have 0km distance? With the dataset, it seems not (using event coordinate and station coordinate).

R: See reply to comment 16.

29) figure 11: "ShakeMap grid" is from Michelini et al. 2020? If yes precise it at least in the figure legend text.

R: We have removed this figure because we have decided to keep a unique column used to assign  $v_{S30}$  at the station locations. See reply to comment 11.

**Referee 2**

General Comments

The publication of an intensity-ground motion dataset is in principle very welcome. It provides a good basis for further research. The dataset also enables better insights into previous studies on empirical relationships between instrumental ground motion data and intensities. The reprint, however, still contains a number of points that require clarification, addition or amendment. Some of these are of a fundamental nature.

Specific Comments

**1 line 7**

It would be important to specify the type of magnitude used for the different earthquakes as part of the dataset. Even the given range can be different if you refer to Mw or ML.

R: We completely agree with the reviewer that this is an important point that the original manuscript required improvement. A similar comment has been raised by Referee 1 (please see also the answers to questions 9, 11, and 15 of Referee1). As explained above, we have decided to adopt only the moment magnitude for all the earthquakes with the aim of providing a clear and unique parameter accompanied by uncertainty. In order to remedy the incompleteness of information about this provided in ESM, we have used the HORUS catalogue, a homogeneous catalog of Italian earthquakes with magnitudes calibrated to Mw. Such homogenized magnitudes were considered for the compilation of the Catalogo Parametrico dei Terremoti Italiani v.2015, which is the data source suggested by the reviewer in one of the following comments.

**2 lines 21-26**

Different DYFI approaches are discussed and the macroseismic scales they use. The following comments arise in this context:

**2-1 Provide the references for the three scales given in lines 25 and 26.**

**2-2 Use the correct reference for the MM scale used for DYFI.**

**2-3 Regarding the use of the MCS scale in Italy, it would be important for readers outside Italy or for readers who are not exactly insiders to know why the MSC scale, which is about 90 years old, is still used in Italy. If**

possible, also give the reference to the authorised translation into Italian, as presumably not all users of the scale in the country can use the original.

**2-4 With regard to the EMS, a distinction must be made between the 1992 and 1998 versions. The EMS-98 version is certainly meant; in this case, the designation must be made accordingly.**

R: With respect to these aspects underlined by the reviewer, we have included references to:

- Wood, H. O., and F. Neumann (1931). Modified Mercalli Intensity Scale of 1931. Bulletin of the Seismological Society of America, 21, 277-283.
- Richter CF (1958) Elementary seismology. Freeman, San Francisco
- Sieberg, A. (1930) Scala mcs (mercalli-cancani-sieberg). Geologie der Erdbeben, Handbuch der Geophysik, 2, 552–555.
- Grünthal G (ed) (1998) European Macroseismic Scale 1998. Cahiers du Centre Europèen de Géodynamique et de Seismologie. Conseil de l'Europe, Conseil de l'Europe.
- Galli, P., Castenetto, S., & Peronace, E. (2017). The macroseismic intensity distribution of the 30 October 2016 earthquake in central Italy (Mw 6.6): Seismotectonic implications. Tectonics, 36, 2179–2191.

**and the following text has been added to Introduction (page 1, line 26)**

"Although since its publication EMS-98 has been widely adopted inside and also outside Europe, MCS scale is still in use in Southern Europe and especially in Italy due to the desire to maintain compatibility with past data sets. Furthermore, as the MCS scale does not fully take into account the vulnerability of each single building, it allows a widespread and expeditious survey, rapidly providing the key information that is directly correlated to the damage level, that is, necessary to the organization of resources for dealing with all humanitarian aspects of the disaster. In turn, as the EMS-98 requires the reconnaissance of the vulnerability class of each building, it is less extensively applicable in the first survey of large earthquakes (Galli et al., 2017)."

**#3 line 54**

What is meant by the slash in the expressions "(PGV)/intensity" and "(SA)/intensity"?

We have replaced "/" with "versus".

**#4 line 57**

It is stated which three spectral acceleration (SA) values are used. On the other hand, chapter 3 deals with practical concerns of the EC8. However, the three used spectral periods are only of limited relevance with regard to the position of the plateau values of the design spectra of the EC8. Further SA values for periods lower than 0.3 s should be added.

Within the context of the manuscript, we aim to provide a data set which correlates strong motion stations and macroseismic intensity observations. In our view, the three periods are actually a good synthesis of the ground motion at short, intermediate and long periods; these three spectral accelerations are chosen to display the amount of shaking experienced by structures sensitive to low periods, intermediate periods, and long periods. Furthermore, they can be easily assumed to be related to PGA, PGV and PGD.

**#5 line 57**

It says: "...we ... remove several earthquakes whose data were proven to be unreliable." It would be important for readers to know which earthquakes this concerns. The knowledge that the authors have gained in this respect should be shared (e.g. in a table with the related earthquakes).

What are the criteria that resulted in excluding certain earthquakes? Are these criteria or findings that have been expressed before or by other authors? Or is this fundamentally new knowledge? A certain transparency would be expected.

Please see our response to referee 1's specific comment 2.

**6 lines 14-65 (Meaning, this comment refers to the introduction in general.) With regard to earlier datasets or works on empirical relations between ground motion data and intensities in Italy, reference is made to Faenza and Michelini (2010, 2011). There are, however, a number of other works specifically for Italy, but also works worth citing for other study areas. Shouldn't these be acknowledged in the introduction? Further down in the text, Gomez-Capera et al. (2020) are mentioned. They refer to at least nine similar papers, five of them concerning Italy. All these papers are based on data sets obviously similar to the one described here by the authors of the reprint.**

R: Thank you for this suggestion. We agree that citing other works in the introduction can be useful to provide a more complete overview of this issue.

We added references to Margottini et al. (1992), Wald et al. (1999), Worden et al. (2012), Zanini et al. (2019) and Masi et al. (2020) when introducing the objective of our work.

Margottini C, Molin D, Serva L (1992) Intensity versus ground motion: a new approach using Italian data. Eng Geol 33:45–58.

Wald DJ, Quitoriano V, Heaton TH, Kanamori H (1999) Relations between peak ground acceleration, peak ground velocity, and modified mercalli intensity in California. Earthq Spectra 15(3):557–564

Worden CB, Gerstenberger MC, Rhoades DA, Wald DJ (2012) Probabilistic relationships between groundmotion parameters and Modified Mercalli intensity in California Bull. Seism Soc Am. 102(1):204–221.

Zanini MA, Hofer L, Faleschini F (2019) Reversible ground motion-to-intensity conversion equations based on the EMS-98 scale. Eng Struct 180:310–320.

Masi A, Chiauzzi L, Nicodemo G, Manfredi V (2020) Correlations between macroseismic intensity estimations and ground motion measures of seismic events. Bull Earthq Eng.

To address the reviewer's comment, in the revised introducion, we now write (page 2, line 52):

"Several studies proposed data sets of different macroseismic intensity scales and ground motion parameters.

In Italy, the first correlations between instrumental parameters and macroseismic intensity scales were defined by Margottini et al. (1992). The authors used a database of 56 records related to 9 Italian earthquakes that occurred between 1980 and 1990. Wald et al. (1999) compared horizontal peak ground motions (PGA and PGV) to observed intensities (MMI) for 8 Californian earthquakes. A large data set of MMI and three ground-motion parameters, such as PGA, PGV and pseudo-spectral acceleration (PSA) deriving from California earthquakes was utilized by Worden et al. (2012).

The starting point of our work has been the data set provided by Faenza and Michelini (2010, 2011). In these works, the authors assembled, respectively, a Peak Ground Acceleration (PGA) and Velocity (PGV) intensity data set and a Spectral Acceleration (SA) intensity data set. They adopted the DBMI04 intensity database (Stucchi et al., 2007), an earlier version of the Italian macroseismic database, and the above mentioned ITACA accelerometric data bank (Pacor et al., 2011).

In the last decade, other authors based their studies on cross-matching of the DataBase of Macroseismic observations of Italy (DBMI) and the ITalian ACceleration Archive (ITACA). Zanini et al. (2019) assembled a PGM versus EMS-98 intensity dataset, collecting 220 data pairs of observations with site-station distances lower than 3 km, from 22 different Italian seismic events. Masi et al. (2020) considered macroseismic data (EMS-98 and MCS scales) and PGMs such as PGA, PGV and Housner Intensity by selecting 179 ground-motion records belonging to 32 earthquake events

occurred in Italy in the last 40 years. Gomez Capera et al. (2020) obtained a dataset that corresponds to 240 pairs of macroseismic intensity-GMPs from 67 Italian earthquakes in the time window 1972–2016 with moment magnitude ranging from 4.2 to 6.8 and macroseismic intensity in the range [2, 10–11].

Here, we merged the latest versions of these sources of data with the inclusion of PGA, PGV and SA values (at 0.3, 1.0, and 3.0 s), we add more recent events and remove several earthquakes whose data were proved to be unreliable, as explained in detail in the following section."

**#7 lines 92-93**

Here it is stated: ". . . the average macroseismic intensity value of a more or less large area with a point." In the following, individual questions on this:

**7-1 Does "more or less large area" mean the locality to which an MDP applies? Please specify.**

**7-2 On the basis of the data set for the reprint https://doi.org/10.13127/inge.1, the localities are villages, small towns and up to medium-sized towns with their geographical extensions. With respect to Ancona it is the centro storico. Right?**

**7-3 What might be meant by "the average macroseismic intensity value"? What are these average intensity values? And from which values was the average or mean calculated? Intensities for a location are determined, as the authors know, from the totality of all macroseismic observations per location - never, as one might think from the wording in the keprint, as the average of point intensities for a location. Only from the totality of all macroseismic observations per location. Only from the totality of all macroseismic observations per location. Only from the totality of all macroseismic observations per location it can be determined in how many cases (or to what percentage) certain effects were observed or not observed. This is because the MCS scale is also based on the use of the frequency (or percentage) of occurrence of certain macroseismic effects per locality, including the types of building damage. The wording in the lines mentioned would therefore have to be corrected or clearly described as to how it was actually done.**

R: Thanks for pointing out this opportunity to clarify the exposition. We have explained these issues in detail in our response to Referee 1's specific comment 1. Furthermore, we have rephrased the mentioned sentence containing "the average macroseismic intensity value of a more or less large area with a point" to read:

"The gazetteer ensures the correspondence between the place name of each locality and a pair of geographical coordinates matching the intensity value representative of the total macroseismic observations per locality with a point (MDP)".

Regarding the comment 7-2, the precise locality associated with the reported observations is Ancona (centro storico).

**#8 line 94.**

**8-1 The intensity is always given in the text, i.e. not only in the line mentioned, with e.g. 6.0, i.e. a decimal number. However, intensities assigned from macroseismic observations are always positive integers from 1 to 12. Therefore, intensities are also indicated with Roman numerals in older literature to emphasise the integer nature of the intensity. However, Roman numerals are somewhat unsuitable for numerical applications. Only derived intensities, i.e. intensities calculated from other quantities by means of empirical relations, appear as decimal numbers. Therefore, either change the intensities to integers or justify why decimals are used.**
**8-2 It is correct to use the notation 6-7 for cases of uncertain macroseismic findings where the intensity can be determined equally well, e.g., as 6 degrees or 7 degrees. But this notation is only used to express the uncertainty described. It is not a range between 6.0 and 7.0, as it says in the mentioned line. It is also by no means an intensity with the accuracy of half a degree. Otherwise the MCS scale would not have 12 degrees**

but 23. The wording would therefore have to be changed in the reprint.

**R: This is a good observation – we agree this is unclear in the original text. These comments are addressed by our reply to comment 19, related to technical corrections of Referee 1.**

**9 line 96**

**9-1 Explain the meaning of the abbreviations "(e.g. "HD", "D", or "F")"; i.e. what do "HD" etc. stand for? #9-2 What means ",class 4.0"? Is "class" used as a synonym for "intensity"? And if so, why?**

**9-3 Aren't classes numbered as natural integers instead of decimals? Please, explain, why you prefer decimals.**

R: We have explained the meaning of the descriptive codes and removed the mentioned sentences about class from the manuscript. The current version is shown and explained in our reply to comment 19, related to technical corrections of Referee 1.

**#10 lines 114-118**

**10-1 It would be extremely helpful for users if a table could be added with the 65 earthquakes that are dealt with here. Here all parameters, such as Mw and ML, can be specified (i.e. both magnitudes!), but also the number of available MDP per event (see lines 133-137). Many of these data are contained in the data set, but would have to be extracted first. A special table for the 65 earthquakes, as described, would be very user-friendly. Please add the number of available MDP per event to the table.**

**10-2 The fact that the reprint does not distinguish whether a magnitude is Mw or ML is, as already mentioned, unacceptable. In addition, the Mw values should be available for all quakes considered in the data set. The CPTI15 file (Rovida et al. 2020; BEE vol. 18) lists all earthquakes in Italy with  $Mw \ge 4.0$  (either true or proxy). The CPTI15 file also contains the related uncertainty for each Mw value. The uncertainty in magnitudes should also be included in the required table.**

R: We agree with the reviewer that the addition of the required table containing the list of the selected seismic events can be useful. The information that was indicated for the characterisation of each earthquake can be summarised as follows: hypocenter with uncertainty, moment magnitude with uncertainty, number of macroseismic data and number of stations for each event (please see the attached Table 1). The comment about the magnitude is addressed by our reply to specific comment 1 above.

**10-3 Why the geographical coordinates of, e.g., M6 earthquakes are given with an accuracy of up to 7 or 8 digits after the decimal point in view of a considerable extension of the fault plane?**

R: The display of up to 7 or 8 digits after the decimal point is due to an error when exporting the table as explained above. We apologize for this mistake in converting file formats. All the data set has been homogenized accounting for a number of decimals consistent with the precision of original measurements.

**#11 lines 119-120**

**11-1 Three columns of  $v_{s30}$  values are given in the data set:  $vs30_m_sec, vs30_m_sec_WA$ ,  $vs30_m_sec_shakemap$ . These designations would also have to be used in the two rows for clarity as to what is meant. Use a capital S, as it comes from S-wave.**

R: To address the reviewer's concern regarding clarity on the meaning of each column heading of the table, we have included a detailed legend in the section dedicated to the description of the dataset on Zenodo, as explained above, to specific comment 1 of Reviewer 1. In order to follow the reviewer's suggestion, in the revised manuscript the average shear-wave velocity for the upper 30-m depth is denoted as  $v_{s30}$ .

**11-2 It would be important to add the uncertainties of the vS30 values. These differ in part significantly for the three types in the data set. In view of the differences in vS30 according to different types of determination at a measuring point of more than 500m/s2 in the data set, this information is more or less useless without explicit error information. If it is not possible to state the uncertainties for all measurement points and all types explicitly, at least summary estimates for certain data groups should be provided for the users of the data set. For future applications, such uncertainty assessments will play an increasing role. In this context, the question arises why the  $v_{s30}$  in m/s are given in the three columns (vs30\_WA, vs30\_shakemap, vs30\_MDP) with an accuracy of up to 6 or 7 digits after the decimal point (!). How does this supposed accuracy relate to the actual uncertainty of these quantities?**

R: We completely agree with the reviewers that the lack of any uncertainty is a key point necessary to discuss. In general, the original ESM database, as well as of the majority of the catalogs cited in the flatfile, do not provide the uncertainty associated with earthquake location parameters (latitude, longitude, and depth), station information (location of the receiver, measured and calculated  $v_{S30}$ ), etc. The inclusion of uncertainty

associated with latitude, longitude, depth and magnitude has been completed through an extensive procedure of search and selection from other catalogs and works present in literature. Regarding this point, please see the answer to the question 9 asked by Reviewer 1. With regard to  $v_{S30}$  values listed in the ESM flatfile, it is not possible to provide uncertainties given the available data.

Furthermore, as explained in our response to comment 10-3, the display of too many digits after the decimal point is due to an error in converting file formats, it is not a direct index of the accuracy of the quantities listed in our data set.

**#12 lines 122**

With regard to the distance between the location for which an intensity value is valid and the measuring point of the strong ground motion station (site\_to\_MDP\_distance\_km), there can only be rough estimates. An intensity value can only be representative for a locality (up to the size of a medium-sized town), which can have a considerable spatial extension. Is the edge of the town then used to determine the distance or its centre, or what? In any case, the uncertainty of the distance parameter also plays a decisive role here. It is essential to specify this uncertainty.

In the published data set, the distance in km is given with 9 digits after the decimal point (!) - and this with regard to a distance from a measuring point to an area of a town with an extension in the order of a few to several kilometres. Is there a reason for the remarkable accuracy of 9 decimal places?

R: The first part of this comment is addressed largely by our reply to Referee 1's specific comment 1. To recap, while we agree with the reviewer's suggestion that the uncertainty of the distance between the MDP and the station should be included, we note that it is not possible to provide a reliable measure of this quantity.

DBMI15 provides the geographical coordinates of the cited localities without indicating whether they represent the centroid of the polygon which corresponds to the area of the locality. Furthermore, it is not specified the spatial extension of each locality. In order to take into account this issue, we have inserted a fixed uncertainty of 0.5 units to the intensity.

Furthermore, as explained in our response to comment 10-3, the display of too many digits after the decimal point is due to an error in converting file formats, it is not a direct index of the accuracy of the quantities listed in our data set.

**13 lines 125-126**

Here it is stated: "... the maximum between the two horizontal components of the peak ground motion measures ...".

**13-1 What do you mean with "the maximum between the two . . . components"?**

Could it be "the maximum difference between the two components"? Or what? Or perhaps "the maximum of the two components"?

**R: We have replaced "the maximum between the two horizontal components" with "the maximum of the two horizontal components" to make the sentence clearer, as suggested.**

**13-2 Measurements of PGA, PGV and calculations of derived vales like SA are connected with uncertainties. These have rarely been used so far, but will be of importance in the future. Since the already published data set covers a relatively large time span from 1972 to 2016, at least qualitative information on uncertainties would be unavoidable.**

It makes a big difference whether data from an early period of strong ground motion measurements are considered (PGA from low dynamic analogue instruments, later somehow digitised) or modern data (PGA measured with 24 bit digitizers). It is obvious that the uncertainty level has changed radically in such a time span. This should be taken into account at least as a discussion.

R: With regards to the reviewer's concern, we are well aware of this issue. Unfortunately, The ESM flatfile table does not contain the uncertainty associated with the peak ground motion measures (PGA and PGV), and the 5% damping elastic response spectral ordinates in acceleration (SA). In order to describe in brief the differences between historical and modern recordings, we have added the following text to Section 2 (page 2, line 68):

Most of ESM data are formed by accelerograms, available through European Integrated Data Archive (EIDA http://www.orfeus-eu.org/eida/eida.html), key infrastructure aimed at archiving digital waveforms. In addition, the ESM database also includes historical data, mainly recorded by analogue instruments.

Digital accelerographs started to become available since the late 90s, and generally operate in continuous mode. The precision of the recorded ground motion depends on the instrument settings, such as digitizer dynamic range, sampling rate and sensor full scale. Data recorded from digital accelerographs have smaller noise to signal as compared to analog accelerographs. This is due to the fact that (a) analog accelerographs are optical mechanical instruments having moving parts (b) these devices generally record ground motion in standby mode and are triggered by a specified acceleration threshold, so they do not preserve the pre- and, sometimes, the post-event time history (c) the natural frequency of transducers and their dynamic range are generally limited (d) it is necessary to digitize the traces in order to be able to use the recording. Due to these reasons, a different treatment of data recorded by analog or digital instruments is automatically implemented when waveforms are uploaded in ESM in order to allow the full compatibility among these recordings (Puglia et al., 2018).

((see full reference here: Puglia R., Russo E., Luzi L., D'Amico M., Felicetta C., Pacor F. and Lanzano G.; 2018: *Strong-Motion processing service: a tool to access and analyse earthquakes strong-motion waveforms*. Bull. Earthquake Eng., 16, 2641-2651.)

**#14 lines 127-129 (macroseismic data)**

**14-1 It is recommended that the already published data set be structured in such a way that, in addition to MCS intensities, those according to EMS-98 can also be included. On the one hand, this would ensure an opening with regard to other parts/countries of Europe where the EMS-98 is routinely used. On the other hand, such an extension could take into account the current developments in Italy to increasingly use the EMS-98 in macroseismology. This is expressed in the fact that more recent earthquakes are increasingly being processed using the EMS-98. Previous earthquakes, such as the 1976 Friuli earthquake of 6 May 1976, have been re-evaluated using the EMS-98 (Tertulliani et al. 2018; BGTA). Recent ground motion-to-intensity conversion equations (i.e., the application domain of the reprint reviewed here including dataset) also use intensity data in Italy in terms of EMS-98 (Zanini et al. 2019; Eng. Struct.) with data from 1983-2016. It is true that, according to Musson et al. (2010), intensity assessments according to MSC and EMS-98 are comparable in principle, but in detail and regarding concrete MDPs, some differences become clearly visible (cf. Tertulliani et al. 2018). So, there would actually be no reason to ignore the described development of also using the EMS-98.**

R: Thank you for this excellent point. We were not aware that despite the macroseismic scale used for most of intensity assessments provided by the Italian macroseismic database (DBMI15) is the MCS scale, also the EMS-98 scale was adopted in some recent surveys whose observations are therein reported. Although, according to Musson et al. 2010, in general terms, the evaluation of the intensities with the MCS scale can reasonably lead to the same intensity value derived from the application of the EMS scale, in order to allow to the user to recognize them, we have added a new column (Int\_type) which contains the type of macroseismic scale reported in DBMI15 for all MDPs selected following the distance criterion.

**14-2  $v_{S30}$  should, according to the original definition, be a point information, i.e. the borehole location at which  $v_{S30}$  was determined. If, on the other hand, a single value is given for  $v_{S30}$  that is representative for the area of a medium-sized town, the error range of this information must not be neglected under any circumstances. This is because in many cases a considerable areal variation of this parameter is observed in the region of such a town. In the data set,  $v_{S30}$  is nevertheless also given here with an accuracy of 6 digits after the decimal point.**

R: We completely agree with the reviewer that this is an important point that the original manuscript was in need of clarification. As explained in our reply to Referee 1's specific comment 1,  $v_{S30}$  at MDPs are punctual information extracted from the  $v_{S30}$  grid adopted by ShakeMap (Michelini et al., 2020) and not directly associated with the macroseismic observations used to determine the intensity of that locality. To provide

uncertainties in this case is really difficult. Moreover, within the DBMI, the extent of the locations is not specified in any way. It is possible to make assumptions; for example, there are 92 localities associated with the ISTAT code of Rome's city. Based on this, it is possible to attribute an average extension to the localities related to Rome's city, and we could do this operation for all the 15343 localities in the DBMI. But this hypothesis is too arbitrary since we know for sure that the localities have different shapes and extensions. Therefore, without any further indications, we thought that any representation concerning  $v_{S30}$  values related to the MDP is too uncertain. For this reason, we have decided not to include this parameter in the data set because it would be likely lead to confusion among users and possible propagation of indeterminate errors. Furthermore, as explained in our response to comment 10-3, the display of too many digits after the decimal point is due to an error in converting file formats, it is not a direct index of the accuracy of the quantities listed in our data set.

**#15 line 133**

Fig. 1 (p. 7) is mentioned here. In the figure, the assigned intensities are given for 3-4 to 10-11. These should therefore be intensity specifications, as it says in line 92, for the cases in which a clear intensity assignment for one degree of intensity in the form of an integer value is not possible. Why are only these uncertain indications shown graphically and not the intensities of the integer values from 4 to 10? These intensities are also more frequent in the data set than the uncertain ones with e.g. 6-7. For reasons of practicability, these can be shown with e.g. 6.5, but it should be clear what we are dealing with such a notation: an uncertain indication as a proxy for 6-7 but in no case an intensity determined exactly to half a degree.

Fig. 1 should be changed accordingly.

**#16 line 135**

Here is referred to Fig. 2. The same applies here as for Fig. 1 (cf. #15).

R: Thank you for the pointer. Both figures have been changed according to reviewer's comment.

**#17 line 177**

It is said: "... small intensity values (i.e. in the range  $3 \le MCS \le 3.5$ )". Here again the at least implicit use of the integer values of intensity as a decimal quantity occurs. There exist in that case only the integer values of 3 and 4. If the observational data are so poor that it can be both 3 or 4 degrees, one writes 3-4. But what does the MCS scale provide for any values in between?

R: The reviewer is correct. We apologize for this mistake. We removed the sentence inside the parenthesis from the manuscript.

**#18 line 186**

A data set with magnitudes of 4.2-6.9 is mentioned here. In the abstract, the data set starts at 4.0. Which is correct?

R: We thank the reviewer for catching the typo. The correct range was 4.0-6.9. After we changed the source of Mw to another in order to include the uncertainty, we have updated this to 4.1-6.8.

**#19 line 187-188**

It is said: " The dataset can be used as reference to benchmark studies seeking correlations between ground motion parameters and MCS macroseismic intensities". That is correct. However, the reader would have liked to see graphs showing exactly these data points. Preferably supplemented by the empirical adjustments based on earlier studies from Faenza & Michelini (2010) to Gomez-Capera et al. (2020); i.e. the graphical representations of the derived empirical relations. This would give a first, albeit only visual, impression of how the new data set behaves. Such a supplement would be very useful.

R: While we think that our data set can be used by other authors to derive new relationships between ground motion parameters macroseismic intensities, we do not agree to address this point in the manuscript. We consider the supplement something to be not included in the paper since the main goal is to present a data set and not to make comparisons with specific studies.

**20 line 198-199**

The use of the data for the determination of Intensity Prediction Equations, as it is called here, should hardly be possible without precisely defined magnitudes in the data set, i.e. not knowing what type of magnitude we are dealing with in individual cases. Compare earlier comments on the specification of magnitude types.

R: The reviewer is right. This has been fixed. Please see our reply to specific comment 1.

**Technical Corrections**

**21 line 67**

"ESM" needs also to be explained in the main body of the text, not only on the abstract.

**22 line 79.**

Mean shear wave velocity of the uppermost 30 m is given as " $V_{S30}$ ". However, the derived physical quantity of velocity is abbreviated with a small v according to ISO (International Organization for Standardization). Unlike PGV, the small v has therefore be used for  $v_{S30}$ . A change according to the ISO standard is recommended.

**23 line 79 and other occurrences below**

With regard to Eurocode 8, it is better not to quote "Code (2005)" but "Eurocode 8 (2005)". Further below, EC8 is used; however, without explanation of the abbreviation.

**24 line 158**

Here it is given "Fig ??b". Insert the appropriate number.

R: All the technical corrections have been done and inserted into the revised paper.

Table 1 List of the selected seismic events: hypocenter with uncertainty, moment magnitude with uncertainty, number of macroseismic data and number of stations for each event are indicated.

| Time                | Lat     | Lon     | Depth | ERH  | ERZ  | Mw   | ERMw | Nb.MDPs | Nb.stations |
|---------------------|---------|---------|-------|------|------|------|------|---------|-------------|
| 1972-06-14 18:55:46 | 43.6880 | 13.4650 | 3.00  |      |      | 4.67 | 0.19 | 2       | 1           |
| 1976-05-06 20:00:12 | 46.2620 | 13.3000 | 5.71  | 1.4  | 1.6  | 6.45 | 0.10 | 6       | 4           |
| 1976-09-11 16:35:01 | 46.2560 | 13.2330 | 4.30  | 1.3  | 1.5  | 5.60 | 0.10 | 1       | 1           |
| 1976-09-15 09:21:18 | 46.3000 | 13.1740 | 11.26 | 0.8  | 0.8  | 5.95 | 0.10 | 2       | 1           |
| 1977-07-24 09:55:30 | 41.1600 | 14.9600 | 35.00 |      |      | 4.19 | 0.19 | 1       | 1           |
| 1977-09-16 23:48:07 | 46.2830 | 13.0190 | 10.78 | 1    | 0.9  | 5.26 | 0.10 | 7       | 5           |
| 1978-03-11 19:20:43 | 38.0500 | 16.0170 | 15.00 | 4    | 3.6  | 5.22 | 0.10 | 5       | 3           |
| 1978-04-15 23:33:47 | 38.4120 | 15.1290 | 17.70 | 3.8  | 3.6  | 6.03 | 0.10 | 8       | 4           |
| 1978-12-05 04:45:26 | 43.0930 | 12.8190 | 10.00 |      |      | 4.30 | 0.19 | 1       | 1           |
| 1979-09-19 21:35:37 | 42.7800 | 13.0000 | 10.00 | 2.8  | 4.8  | 5.83 | 0.10 | 15      | 5           |
| 1980-01-05 14:32:26 | 45.0510 | 7.3680  | 15.00 | 3.5  | 7.7  | 4.82 | 0.10 | 2       | 1           |
| 1980-02-20 02:34:01 | 39.2900 | 16.1500 | 3.70  |      |      | 4.42 | 0.10 | 2       | 1           |
| 1980-02-28 21:04:40 | 42.7530 | 12.9960 | 12.90 | 3.1  | 4.8  | 4.97 | 0.10 | 1       | 1           |
| 1980-06-07 18:35:01 | 44.0500 | 10.6000 | 30.00 |      |      | 4.64 | 0.10 | 7       | 3           |
| 1980-06-09 16:02:47 | 42.1860 | 13.7810 | 39.30 |      |      | 4.64 | 0.10 | 1       | 1           |
| 1980-11-23 18:34:53 | 40.8700 | 15.3780 | 10.00 | 3.7  | 3.3  | 6.81 | 0.10 | 21      | 17          |
| 1980-12-09 05:50:12 | 38.7600 | 16.1810 | 55.00 | 6.5  | 19.8 | 4.67 | 0.10 | 4       | 3           |
| 1981-06-07 13:01:00 | 37.6740 | 12.4770 | 21.40 |      |      | 4.93 | 0.10 | 1       | 1           |
| 1982-03-21 09:44:00 | 39,7043 | 15.6385 | 18.90 | 2.3  | 0.7  | 5.23 | 0.10 | 1       | 1           |
| 1983-07-20 22:03:30 | 37.5487 | 15,1680 | 24.70 | 2    | 1.6  | 4.10 | 0.50 | 6       | 1           |
| 1983-11-09 16:29:52 | 44.6487 | 10.3665 | 28.10 | 0.1  | 0.1  | 5.04 | 0.10 | 3       | 1           |
| 1984-04-29 05:03:00 | 43.2100 | 12.5700 | 5.97  | 0.1  | 0.8  | 5.62 | 0.10 | 7       | 5           |
| 1984-05-07 17:49:43 | 41.7000 | 13.8600 | 20.50 | 0.1  | 0.1  | 5.86 | 0.10 | 16      | 10          |
| 1984-05-11 10:41:48 | 41.7800 | 13.8900 | 12.10 | 0.1  | 0.2  | 5.47 | 0.10 | 5       | 5           |
| 1987-05-02 20:43:54 | 44,7940 | 10.6780 | 23.67 | 0.1  | 0.1  | 4.71 | 0.10 | 2       | 1           |
| 1988-02-01 14:21:40 | 46.3590 | 13.0750 | 3.10  | 0.2  | 0.4  | 4.94 | 0.21 | 5       | 5           |
| 1990-12-13 00:24:26 | 37.3300 | 15.2410 | 0.31  | 0.7  | 9.1  | 5.61 | 0.10 | 7       | 6           |
| 1995-10-10 06:54:22 | 44.1330 | 10.0180 | 8.23  | 0.3  | 0.7  | 4.82 | 0.10 | 1       | 1           |
| 1996-10-15 09:56:00 | 44.7630 | 10.6050 | 25.54 | 0.3  | 0.3  | 5.38 | 0.10 | 3       | 2           |
| 1997-09-03 22:07:30 | 43.0260 | 12.8770 | 5.74  | 0.1  | 0.4  | 4.54 | 0.07 | 4       | 2           |
| 1997-09-26 09:40:24 | 43.0150 | 12.8540 | 9.87  | 0.1  | 0.3  | 5.97 | 0.07 | 21      | 15          |
| 1998-09-09 11:28:00 | 40.0600 | 15.9490 | 29.21 | 0.7  | 0.3  | 5.53 | 0.07 | 2       | 2           |
| 1999-02-14 11:45:53 | 38.2660 | 15.0220 | 20.67 | 0.2  | 0.2  | 4.66 | 0.07 | 3       | 1           |
| 2001-04-22 13:56:34 | 37.7230 | 14.9890 | 0.03  | 0.2  | 1.7  | 4.19 | 0.07 | 1       | 1           |
| 2002-04-05 04:52:21 | 39.1660 | 15.4800 | 0.00  | 0.4  | 2.1  | 4.49 | 0.07 | 1       | 1           |
| 2002-09-06 01:21:28 | 38.3810 | 13.6540 | 27.01 | 0.4  | 0.4  | 5.91 | 0.07 | 3       | 3           |
| 2002-10-27 02:50:26 | 37.7660 | 15.1060 | 0.04  | 0.3  | 7    | 4.84 | 0.07 | 1       | 1           |
| 2003-01-26 19:57:03 | 43.8830 | 11.9600 | 6.53  | 1.77 | 1.46 | 4.67 | 0.07 | 3       | 2           |
| 2003-04-11 09:26:57 | 44.7580 | 8.8680  | 8.15  | 1.49 | 5.08 | 4.81 | 0.07 | 1       | 1           |
| 2003-09-14 21:42:53 | 44.2550 | 11.3800 | 8.33  | 1.64 | 2.52 | 5.24 | 0.07 | 2       | 2           |
| 2004-11-24 22:59:38 | 45.6850 | 10.5210 | 5.44  | 1.1  | 0.77 | 4.99 | 0.07 | 2       | 1           |
| 2006-02-27 04:34:01 | 38.1550 | 15.2000 | 9.20  | 1.06 | 1.2  | 4.38 | 0.07 | 7       | 6           |
| 2006-12-19 14:58:06 | 37.7780 | 14.9130 | 23.80 | 1.14 | 1.2  | 4.20 | 0.07 | 2       | 2           |
| 2008-12-23 15:24:21 | 44.5440 | 10.3450 | 22.90 | 1.06 | 0.9  | 5.36 | 0.07 | 6       | 6           |
| 2009-04-06 01:32:40 | 42.3420 | 13.3800 | 8.30  | 0.71 |      | 6.29 | 0.07 | 47      | 13          |
| 2009-11-08 06:51:16 | 37.8470 | 14.5570 | 7.60  | 0.99 | 1.2  | 4.52 | 0.07 | 1       | 1           |
| 2009-12-15 13:11:58 | 43.0070 | 12.2710 | 8.80  | 0.71 | 1    | 4.22 | 0.07 | 1       | 1           |
| 2009-12-19 09:01:16 | 37.7820 | 14.9740 | 26.90 | 1.28 | 1.4  | 4.40 | 0.07 | 7       | 6           |

| 2010-04-02 20:04:45 | 37.7990 | 15.0790 | 0.31  | 0.2  | 0.2 | 4.20 | 0.07 | 1   | 1  |
|---------------------|---------|---------|-------|------|-----|------|------|-----|----|
| 2010-08-16 12:54:47 | 38.4100 | 14.9190 | 16.90 | 9.22 | 1   | 4.68 | 0.07 | 4   | 3  |
| 2011-05-06 15:12:35 | 37.8040 | 14.9430 | 20.35 | 0.3  | 0.5 | 4.30 | 0.07 | 1   | 1  |
| 2011-06-23 22:02:46 | 38.0640 | 14.7840 | 7.30  | 0.92 | 1.1 | 4.70 | 0.07 | 7   | 7  |
| 2011-07-17 18:30:27 | 45.0100 | 11.3670 | 2.40  | 0.94 |     | 4.68 | 0.07 | 4   | 4  |
| 2011-07-25 12:31:20 | 45.0160 | 7.3650  | 11.00 | 1.39 |     | 4.55 | 0.07 | 4   | 4  |
| 2012-01-25 08:06:37 | 44.8710 | 10.5100 | 29.00 | 0.86 | 0.7 | 4.98 | 0.07 | 3   | 3  |
| 2012-05-20 02:03:50 | 44.8955 | 11.2635 | 9.50  | 0.72 | 1   | 6.09 | 0.07 | 2   | 1  |
| 2012-10-25 23:05:24 | 39.8747 | 16.0158 | 9.70  | 0.64 | 0.7 | 5.32 | 0.07 | 15  | 14 |
| 2013-01-04 07:50:06 | 37.8810 | 14.7190 | 9.57  | 0.3  | 0.3 | 4.37 | 0.07 | 2   | 2  |
| 2013-06-21 10:33:56 | 44.1308 | 10.1357 | 7.00  | 0.91 |     | 5.32 | 0.07 | 8   | 3  |
| 2013-08-15 23:06:51 | 38.1627 | 14.9138 | 24.80 | 0.75 | 0.9 | 4.27 | 0.19 | 6   | 5  |
| 2013-12-29 17:08:43 | 41.3952 | 14.4342 | 20.40 | 0.37 | 0.6 | 5.14 | 0.07 | 7   | 5  |
| 2016-02-08 15:35:43 | 36.9745 | 14.8678 | 7.40  | 0.83 | 0.8 | 4.43 | 0.07 | 11  | 10 |
| 2016-08-24 01:36:32 | 42.6983 | 13.2335 | 8.10  | 0.15 | 0.2 | 6.18 | 0.07 | 49  | 26 |
| 2016-10-26 19:18:06 | 42.9048 | 13.0902 | 9.60  | 0.2  | 0.2 | 6.08 | 0.07 | 23  | 20 |
| 2016-10-30 06:40:18 | 42.8303 | 13.1092 | 10.00 | 0.19 | 0.2 | 6.61 | 0.07 | 114 | 55 |